# Structural insights into the committed step of bacterial phospholipid biosynthesis

Zhenjian Li[1], Yannan Tang [1,2], Yiran Wu [3], Suwen Zhao [3,4], Juan Bao[1], Yitian Luo [1,2,4] & Dianfan Li [1]

The membrane-integral glycerol 3-phosphate (G3P) acyltransferase PlsY catalyses the committed and essential step in bacterial phospholipid biosynthesis by acylation of G3P, forming lysophosphatidic acid. It contains no known acyltransferase motifs, lacks eukaryotic homologs, and uses the unusual acyl-phosphate as acyl donor, as opposed to acyl-CoA or acyl-carrier protein for other acyltransferases. Previous studies have identified several PlsY inhibitors as potential antimicrobials. Here we determine the crystal structure of PlsY at 1.48 Å resolution, revealing a seven-transmembrane helix fold. Four additional substrate- and product-bound structures uncover the atomic details of its relatively inflexible active site. Structure and mutagenesis suggest a different acylation mechanism of 'substrate-assisted catalysis' that, unlike other acyltransferases, does not require a proteinaceous catalytic base to complete. The structure data and a high-throughput enzymatic assay developed in this work should prove useful for virtual and experimental screening of inhibitors against this vital bacterial enzyme.

[1] State Key Laboratory of Molecular Biology, National Center for Protein Science Shanghai, Shanghai Science Research Center, CAS Center for Excellence in Molecular Cell Science, Shanghai Institute of Biochemistry and Cell Biology, Chinese Academy of Sciences, 333 Haike Road, Shanghai 201210, China. [2] University of Chinese Academy of Sciences, , Shanghai, 201210, China. [3] iHuman Institute, ShanghaiTech University, 333 Middle Huaxia Road, Shanghai 201210, China. [4] School of Life Science and Technology, ShanghaiTech University, 333 Middle Huaxia Road, Shanghai 201210, China. Zhenjian Li and Yannan Tang contributed equally to this work. Correspondence and requests for materials should be addressed to D.L. (email: dianfan.li@sibcb.ac.cn)

Phospholipids constitute a major component of various membranes in all domains of life. Phospholipid biosynthesis begins with acylation of glycerol 3-phosphate (G3P) to form lysophosphatidic acid (lysoPA), which undergoes a second acylation to form phosphatidic acid (PA), a central intermediate in phospholipid metabolism[1–4]. In bacteria, two distinct G3P acyltransferases (GPATs), PlsB and PlsY, catalyse the committed step. The conventional PlsB has eukaryotic homologs and uses thioesters (acyl-CoA or acyl-carrier protein) as the acyl donor. By contrast, the recently identified membrane-integral PlsY exists exclusively and ubiquitously in bacteria, and uses an unusual acyl donor, namely the acyl-phosphate (acylP)[5], representing a unique class of acyltransferase. It is the sole and hence essential GPAT in most Gram-positive bacteria[1,5–7] such as *Enterococcus faecium* and *Streptococcus pneumoniae*, identified by the World Health Organization as the most dangerous multi-drug resistant pathogens. In the Gram-negative bacterium *Escherichia coli* that contains both GPATs (PlsB and PlsY), deletion of both PlsY and the acylP-synthesizing enzyme PlsX is lethal[7], suggesting the essentiality of the PlsX/PlsY pathway. Given the medical importance of PlsY, previous studies have synthesized and screened a panel of acylP analogous, and identified several acyl-sulfamates as potential PlsY-inhibiting antimicrobials for *Staphylococcus aureus*[8–10]. Such drug-discovery studies would benefit from virtual screening, where large libraries of small molecules are assessed relatively quickly and inexpensively *in silico*, compared with the experimental high-throughput screening approach[11]. However, virtual screening relies on high-resolution structures to provide atomic details of regions of interest, which are generally the active site or allosteric regulation sites of target proteins[11].

PlsY is relatively small (~ 200 residues) but extremely hydrophobic, a characteristic that might have hindered its structure determination. Previous studies have identified five transmembrane helices (TMHs) in *S. pneumoniae* PlsY (*sp*PlsY) using the substituted Cys-accessibility method (SCAM) and deduced three functionally important motifs for acylP binding (residues 35–46), G3P binding (residues 100–107), and catalysis (residues 185–197), using sequence alignment and mutagenesis[1,12]. The detailed architecture of the active site and enzyme–ligand interaction mode awaits for structural characterization of PlsY in complex with its substrates and/or products.

PlsY contains a catalytically important residue His185 (*sp*PlsY numbering). This has led to the hypothesis that PlsY shares a similar catalytic mechanism to the 'Asp-His' dyad mechanism for its thioester counterparts[1,12]. In the $HX_4D$ motif common to these conventional acyltransferases, the aspartate residue raises the $pK_a$ of the histidine for deprotonating the G3P 1-OH to launch nucleophilic attack on the thioester, forming lysoPA[13]. However, such a motif is not found in PlsY. This, together with other aforementioned unique features in protein sequence and acyl substrates, indicates that PlsY may carry out the acylation reaction through a mechanism that is distinctly different from the dyad model. This hypothesis needs to be tested by structural and functional studies of PlsY.

Here we use X-ray crystallography to determine the PlsY structure at 1.48 Å resolution using crystals grown in an activity-supporting lipid bilayer environment, revealing a 7-TMH fold. We also obtain four additional high-resolution PlsY structures, collectively with all the substrates and products bound. The liganded structures, along with extensive mutagenesis data,

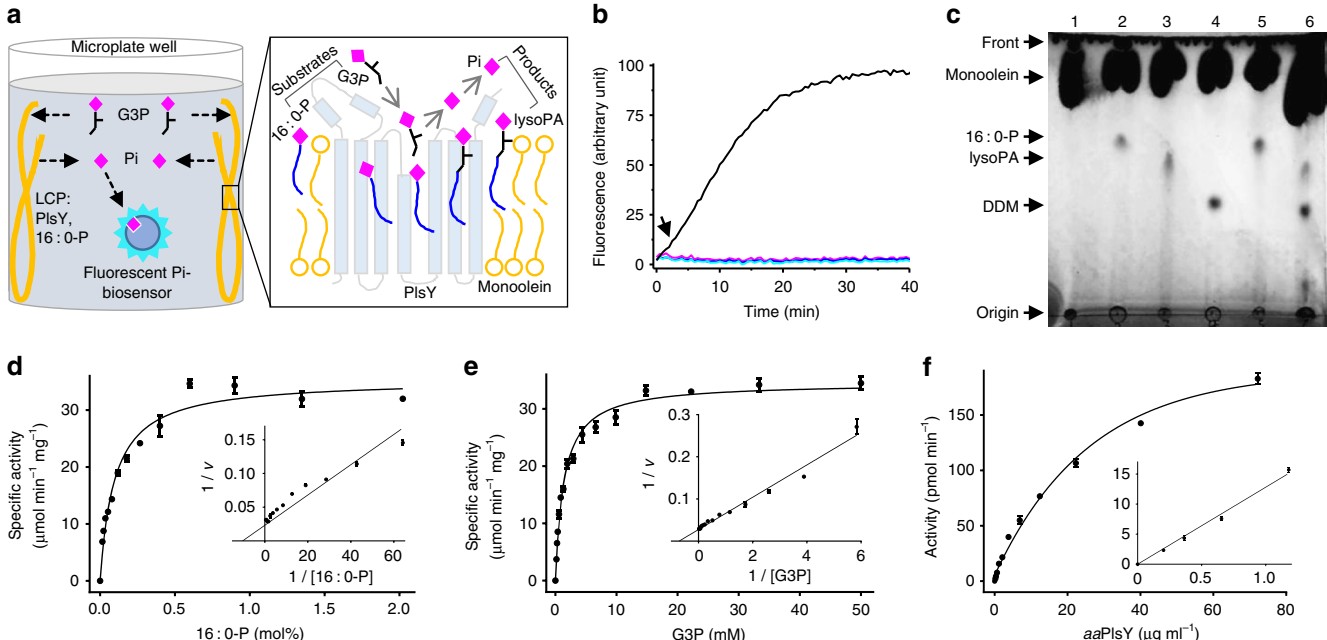

**Fig. 1** Characterization of *aa*PlsY in LCP. **a** The principle of the Pi-biosensor assay. The sticky LCP loaded with *aa*PlsY and its substrate 16:0-P is dispensed at the side of a microplate well and is repeatedly soaked to remove contaminant Pi. G3P, pre-treated enzymatically with a Pi-scavenger[14], diffuses from the bathing solution to LCP, and initiates the reaction. The product Pi is then released from LCP to the bathing solution, triggering fluorescence increase of the Pi-biosensor[14], which is monitored continuously using a plate reader. **b** Progress curves of the coupled assay omitting G3P (magenta), 16:0-P (cyan), enzyme (blue), and none of the above (black). The 2 min lag phase (indicated by an arrow) reflects the time required for equilibration of G3P between the bathing solution and LCP[16]; the linear phase is used to calculate activity. **c** Thin layer chromatography assay of *aa*PlsY activity. Lane 1, monoolein; lane 2, monoolein with 16:0-P (substrate); lane 3, monoolein with lysoPA (product); lane 4, monoolein with *n*-dodecyl-β-ᴅ-maltoside; lane 5, reaction without *aa*PlsY; lane 6, reaction with *aa*PlsY. The brightness and contrast of the whole image have been adjusted to enhance the substrate and product bands. **d**, **e** Michaelis–Menten and Lineweaver–Burk plot (insets) of kinetic measurements for the indicated substrates. **f** Dependence of activity rate on enzyme loading. In **d–f**, data represent the average ± SD from triplicate measurements

**Table 1 Data collection and refinement statistics**

| | $aa$PlsY$_{MAG}$ | SeMet-$aa$PlsY | $aa$PlsY$_{16:0\text{-}P}$ | $aa$PlsY$_{G3P}$ | $aa$PlsY$_{lysoPA}$ | $aa$PlsY$_{Pi}$ |
|---|---|---|---|---|---|---|
| *Data collection* | | | | | | |
| Space group | I222 | I222 | P2$_1$2$_1$2$_1$ | P22$_1$2$_1$ | C2 | C222$_1$ |
| *Cell dimensions* | | | | | | |
| $a, b, c$ (Å) | 57.68, 83.44, 107.59 | 57.68, 83.44, 107.42 | 46.22, 65.60, 84.68 | 42.15, 53.77, 87.69 | 123.89, 40.90, 91.71 | 58.34, 97.99, 83.31 |
| $\alpha, \beta, \gamma$ (°) | 90, 90, 90 | 90, 90, 90 | 90, 90, 90 | 90, 90, 90 | 90, 95.59, 90 | 90, 90, 90 |
| Wavelength (Å) | 0.97852 | 0.97861 | 0.97854 | 0.97853 | 0.97915 | 0.97915 |
| Resolution (Å) | 47.44–1.48 (1.51–1.48)$^a$ | 47.45–2.04 (2.09–2.04) | 46.22–1.77 (1.81–1.77) | 45.84–2.37 (2.46–2.37) | 35.33–2.41 (2.50–2.41) | 31.73–1.83 (1.89–1.83) |
| $R_{merge}$ | 0.054 (0.893) | 0.173 (1.245) | 0.064 (0.798) | 0.173 (1.145) | 0.099 (0.750) | 0.175 (0.889) |
| $R_{pim}$ | 0.025 (0.428) | 0.060 (0.450) | 0.029 (0.429) | 0.052 (0.382) | 0.060 (0.448) | 0.075 (0.433) |
| $I/\sigma I$ | 16.6 (1.7) | 10.9 (2.0) | 18.0 (2.0) | 11.9 (1.7) | 11.37 (2.0) | 7.02 (2.28) |
| Completeness (%) | 98.4 (94.4) | 99.7 (96.4) | 98.4 (97.1) | 98.9 (89.9) | 99.1 (96.8) | 95.35 (86.02) |
| Multiplicity | 5.6 (5.1) | 9.5 (8.7) | 5.6 (5.2) | 12.4 (9.7) | 3.7 (3.7) | 6.0 (5.1) |
| $CC^{\star}$ $^b$ | 0.998 (0.916) | 0.995 (0.856) | 0.999 (0.914) | 0.999 (0.917) | 0.998 (0.958) | 0.997 (0.939) |
| *Refinement* | | | | | | |
| Resolution (Å) | 35.58–1.48 | | 40.57–1.77 | 45.84–2.37 | 35.33–2.41 | 31.73–1.83 |
| No. reflections | 42,709 | | 25,262 | 8,437 | 17,846 | 20,554 |
| $R_{work}$ / $R_{free}$ | 0.1877/0.2128 | | 0.1954/0.2254 | 0.2187/0.2631 | 0.2311/0.2517 | 0.2055/0.2500 |
| No. atoms | 2,012 | | 1,785 | 1,690 | 3,119 | 1,874 |
| Protein | 1,559 | | 1,522 | 1,559 | 3,060 | 1,569 |
| Ligand/ion | 380 | | 210 | 115 | 42 | 201 |
| Water | 73 | | 53 | 16 | 17 | 104 |
| No. residues | 199 | | 193 | 200 | 395 | 200 |
| B-factors (Å$^2$) | 27.53 | | 31.84 | 43.78 | 46.46 | 18.18 |
| Protein | 22.70 | | 29.31 | 42.92 | 46.34 | 15.77 |
| Ligand/ion | 46.00 | | 49.25 | 55.23 | 54.51 | 32.46 |
| Water | 34.42 | | 35.34 | 45.23 | 47.06 | 26.88 |
| *R.m.s deviations* | | | | | | |
| Bond lengths (Å) | 0.007 | | 0.007 | 0.097 | 0.002 | 0.007 |
| Bond angles (°) | 0.760 | | 0.770 | 1.72 | 0.460 | 0.800 |
| *Ramachandran* | | | | | | |
| Favored (%) | 100 | | 100 | 98.99 | 98.20 | 100 |
| Allowed (%) | 0 | | 0.0 | 1.01 | 1.80 | 0 |
| Outlier (%) | 0 | | 0 | 0 | 0 | 0 |
| **PDB ID** | 5XJ5 | | 5XJ7 | 5XJ6 | 5XJ8 | 5XJ9 |

$^a$Highest resolution shell is shown in parenthesis
$^b CC^{\star} = \sqrt{\frac{2CC_{1/2}}{1+CC_{1/2}}}$

suggest a different 'substrate-assisted catalysis' mechanism, through which the acylation of G3P is accomplished without a catalytic base from the enzyme.

## Results

**A high-throughput PlsY assay in a lipid environment.** To identify stable protein for crystallization, we screened five orthologs and selected the $aa$PlsY from *Aquifex aeolicus*. This hyperthermophile-origin protein shares 37% identity and 55% similarity with the functionally characterized $sp$PlsY[5] (Supplementary Fig. 1a), and contains all the functionally critical residues[12]. To positively identify $aa$PlsY as a GPAT, biochemical assays were designed and performed as below.

PlsY acylates G3P using acylP, forming lysoPA and inorganic phosphate (Pi) (Supplementary Fig. 1b). The Pi-releasing activity can be measured using a fluorophore-labeled Pi-biosensor[14], of which the fluorescence increases upon binding of Pi. First, we performed the assay using detergent-solubilized $aa$PlsY premixed with G3P and the Pi-biosensor. Palmitoyl phosphate (16 : 0-P) was added to initialize the reaction. However, the fluorescence of the Pi-biosensor was always saturated by contaminant Pi in the assay mixture, either from the protein purification, or the synthesis and degradation of the unstable 16 : 0-P. To overcome this problem, we developed a lipid cubic phase (LCP)-based assay

through which the contaminant Pi can be removed conveniently by simple soaking before the assay, as outlined below.

LCP is a bi-continuous material containing a lipid bilayer that separates two water channels[15]. It has a gel-like texture and is viscous, sticky, and nano-porous. For the coupled assay, purified $aa$PlsY (Supplementary Fig. 2a, 2b) and acylP were reconstituted into LCP, which was then deposited on the sidewall of microplate wells to avoid interference for later spectroscopic measurements. Pi-free buffer was added to soak the LCP. During the soaking, the sticky LCP remains in the sidewall, retaining the hydrophobic PlsY and acylP in its bilayer. By contrast, water-soluble Pi contaminants diffuse from LCP water channels to the soaking buffer and are therefore removed. Upon addition of G3P, this small water-soluble substrate diffuses into LCP and initiates the reaction, producing lysoPA and Pi. Pi is released from LCP to the soaking Assay Mix, increasing fluorescence of the pre-loaded Pi-biosensor (Fig. 1a).

As shown in the progress curves (Fig. 1b), the fluorescence of the Pi-biosensor increases as a function of time, in a substrate- and $aa$PlsY-dependant manner. An initial lag phase reflects the time required for equilibrium of G3P between the Assay Mix and LCP. The lag phase is ~ 2 min, much shorter than that for the diacylglycerol kinase (DgkA)[16], presumably because the diffusion rate of G3P (172.07 g mol$^{-1}$) is much higher than the DgkA's substrate ATP (507.18 g mol$^{-1}$) due to its smaller size. The lag

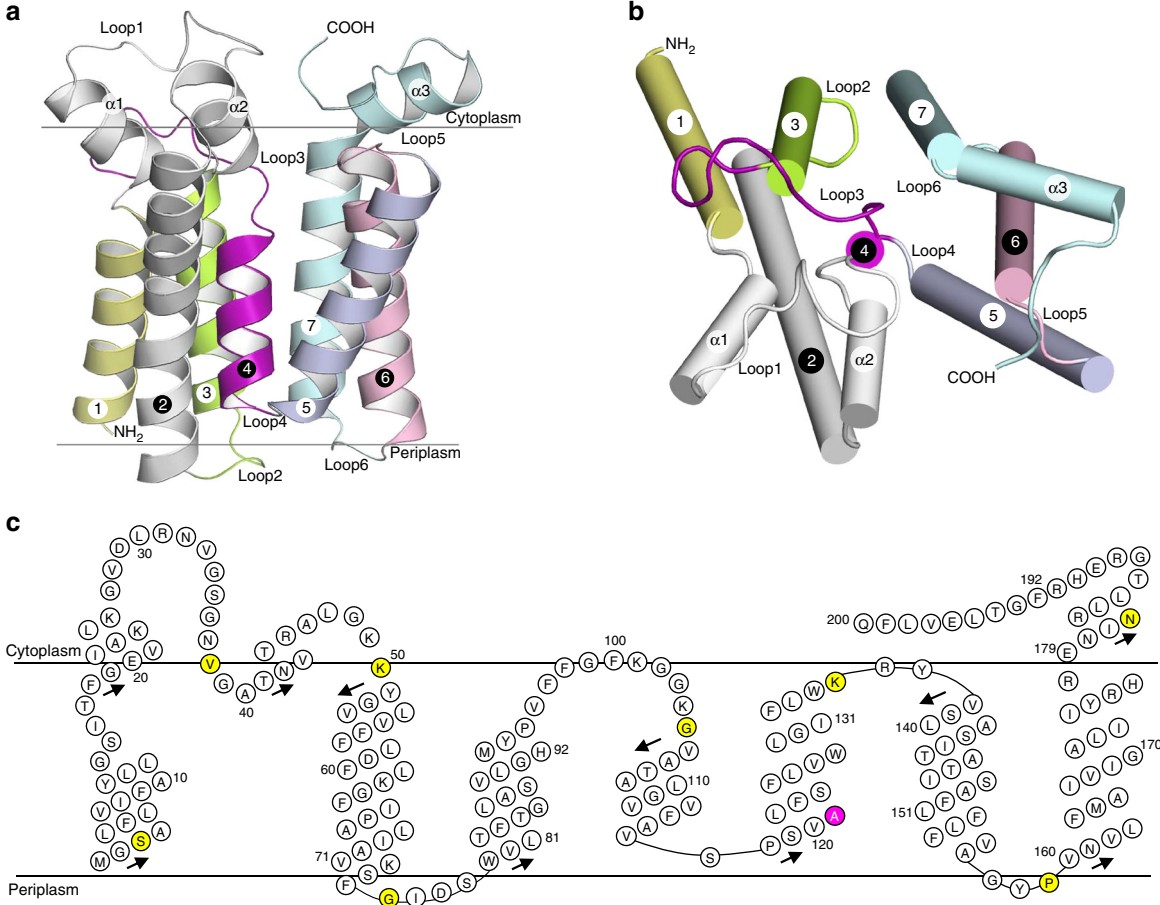

**Fig. 2** Crystal structure and membrane topology of *aa*PlsY. **a** View from the membrane plane. The membrane boundary was defined by the OPM server[68]. TMHs and loops are appropriately labeled. **b** View from the cytoplasm. **c** Membrane topology of *aa*PlsY. Residues with consistent topological assignment between the previous report[12] and our structure are highlighted in yellow; the single outlier (Ala121) is highlighted in magenta. Arrows mark helix directions from N to C termini

phase is followed by a linear phase, which was used to calculate activity.

As a validation, the production of lysoPA was assayed using thin layer chromatography (TLC). As shown in Fig. 1c, lysoPA was detected in the LCP sample loaded with *aa*PlsY, but not in the absence of the enzyme.

Using the coupled assay, the dependence of *aa*PlsY activity on substrate concentration was investigated. *aa*PlsY showed classic Michaelis–Menten kinetics with $V_{max}$ of 34.55 μmol min$^{-1}$ mg$^{-1}$ and apparent $K_m$ of 0.1 mol% palmitoyl phosphate (16 : 0-P, relative to monoolein, the LCP lipid) (Fig. 1d), and 1.39 mM G3P (Fig. 1e). Further, the activity rate of *aa*PlsY depended on the enzyme concentration (Fig. 1f). These results suggest that *aa*PlsY is a well-behaved enzyme in LCP and that LCP is a functionally relevant membrane mimetic for *aa*PlsY crystallization.

**PlsY adapts a 7-TMH membrane topology**. To define the overall architecture of *aa*PlsY, we performed protein crystallization in both detergent micelles and LCP. Despite its ultra-thermostability (Supplementary Fig. 2c), crystallization trials in *n*-decyl-β-D-maltoside or *n*-octyl-β-D-glucopyranoside detergents did not yield protein crystals. By contrast, rod-shaped *aa*PlsY crystals were obtained in LCP made of 7.8 monoacylglycerol (MAG) (Supplementary Fig. 2d), a powerful host lipid for several crystallization projects[17–19]. The structure was solved with selenomethioninyl crystals using single-wavelength anomalous dispersion[20] (Table 1), at a high resolution of 1.48 Å.

The structure (named as *aa*PlsY$_{MAG}$) contains one complete *aa*PlsY monomer in each asymmetric unit. The overall shape of *aa*PlsY resembles a funnel composed of three tilted helices (α1–3) sitting in a cup composed of seven TMHs (Fig. 2a). The TMHs are uniquely arranged compared with six known 7-TMH folds (Supplementary Fig. 3)[21–26]. The membrane-perpendicular TMH4 is sandwiched by two bundles, TMH1–3 and TMH5–7 (Fig. 2a, b). The striking shortness of TMH4 ($n = 11$, Fig. 2c) is compensated by the membrane-embedded Loop3. The overall structure is very compact; apart from two relatively long cytoplasmic loops containing functionally important residues, the helices are stitched together with short linkers.

**The V-shaped active site**. Mapping functionally important residues[12] onto *aa*PlsY$_{MAG}$ reveals a V-shaped cavity that hosts two $SO_4^{2-}$ and a MAG (Fig. 3a) with well-defined electron density (Supplementary Fig. 4a). It is noticed that the MAG and $SO_4^{2-}/1$ are positioned in a way that mimics the lipid substrate acylP (Supplementary Fig. 4a). As well, the $SO_4^{2-}/2$ at the membrane interface should mimic the phosphate group of G3P. Therefore, this cavity is speculated to be the active site. This is confirmed by four additional structures; two with substrates (*aa*PlsY$_{16:0-P}$, 1.77 Å; *aa*PlsY$_{G3P}$, 2.37 Å) (Table 1 and Supplementary Fig. 4b, 4c), and two with products (*aa*PlsY$_{Pi}$, 1.83 Å; *aa*PlsY$_{lysoPA}$, 2.41 Å) (Table 1 and Supplementary Fig. 4d, 4e). All ligands align well in the active site (Fig. 3b). Despite the different states, the five structures are superimposable with root-mean-square deviation

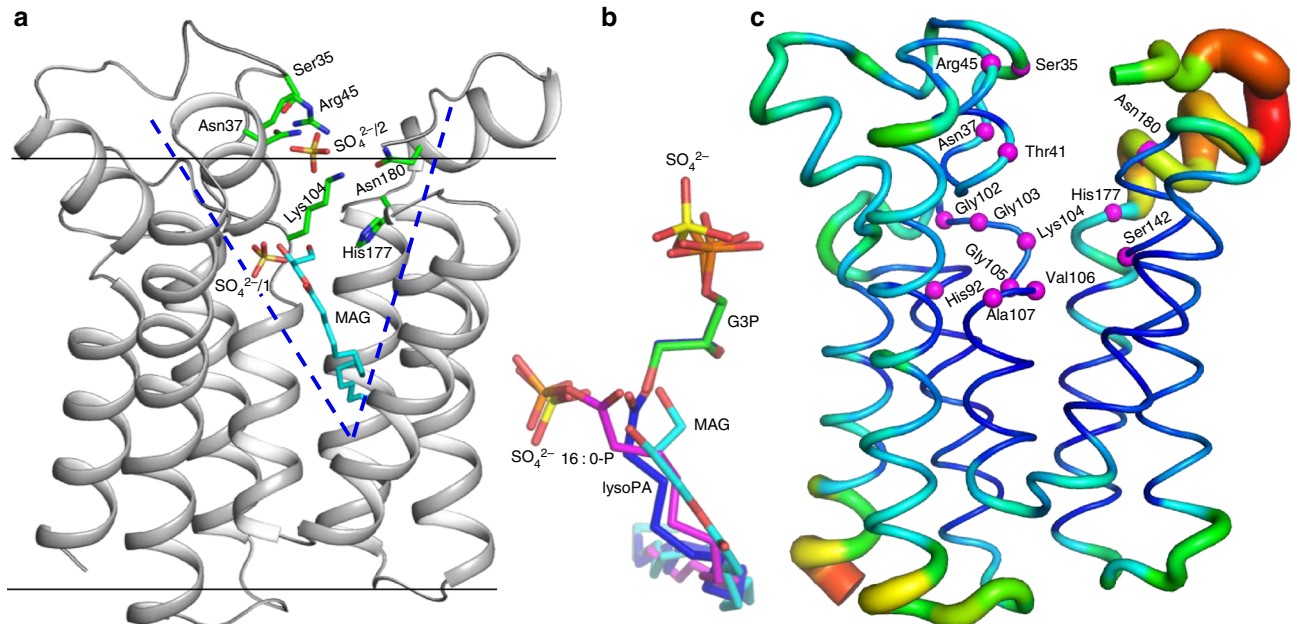

**Fig. 3** The active site of *aa*PlsY. **a** The V-shaped active site hosting a MAG molecule and two $SO_4^{2-}$ ions. Active site residues from a previous report[12], the MAG, and the two sulphates were shown in sticks. **b** Superimposition of all active site ligands. The MAG, 16:0-P, G3P, and lysoPA are shown in stick representations with the oxygen atoms in red, sulphur atoms in yellow, and carbon atoms in cyan, magenta, green, and blue, respectively. **c** Putty representation of B-factors of the *aa*PlsY$_{MAG}$ structure. Blue lines and red tubes indicate low and high B-factors, respectively. The Cα atoms of active site residues are highlighted as magenta spheres

values within 0.52 Å (Supplementary Fig. 5a), suggesting a lack of flexibility. Indeed, all substrate/product-interacting residues except Asn180 showed low B-factors (Fig. 3c), suggesting a relatively inflexible active site, an attractive feature for *in silico* inhibitor screening.

**The phosphate hole for the lipid substrate acylP.** *aa*PlsY is small (192 residues) and monomeric; yet it crosses membrane seven times and contains pockets that must host two substrates with distinct differences in solubility. This implies an efficient structure build for function. Analysing the 16:0-P binding details reveals how this is achieved. Structure elements such as backbone amides and TMHs are the major components of the binding site (Fig. 4a–c); as well, some residues are either involved in the binding of both substrates or involved in 16:0-P binding via both backbone amides and side chains (Fig. 4a, d).

In detail, backbone amides from four consecutive residues (Lys104, Gly105, Val106, and Ala107) in Loop3, the side chain of His 92 (Fig. 4a), and two α-helix dipoles, namely the partially positively charged N termini of TMH4 and α2 (Fig. 4b), form a hydrophilic 'phosphate hole' deep in the membrane (Fig. 4a, b), securing the phosphate group of 16:0-P. Two backbone amides from Ala40 and Thr41, and the Thr41 hydroxyl, form three H-bonds with the carbonyl. Three water molecules further stabilize the acylP head group in this hydrophilic interior. As for the hydrophobic portion of 16:0-P, the structure scaffold components again, namely the TMH2 and TMH4-6, form a hydrophobic groove hosting the palmitoyl chain (Fig. 4c). Notably, the 'phosphate hole' component Lys104 also forms charge–charge interaction with G3P via its side chain amine (see below), supporting the notion that the binding site was efficiently built by making the maximum use of functional groups and structural elements.

As noted, His92 and Thr41 are the only two residues that contribute to acylP binding by side chains. Mutating Thr41 to alanine had severe consequence in catalytic activity (Table 2 and

Fig. 4e), whereas keeping the hydroxyl by serine mutation only reduced half of the activity. Mutation of His92 to alanine also caused a dramatic 80% loss of activity. The charge property and spatial arrangement of His92 appear to be optimal because extensive mutagenesis failed to find a competent substitution (Table 2 and Fig. 4e). Our mutagenesis data also show that the distortion of the phosphate hole by proline mutations (G105P and V106P) almost abolished the acylation activity (Table 2 and Fig. 4e), in agreement with the structure observations.

**The invariable G3P binding pocket.** The *aa*PlsY$_{G3P}$ structure reveals a G3P pocket composed of a 'phosphate clamp' (Ser35, Arg45, Lys104, and Ans180), which binds the phosphate of G3P, and two side chains (Ser142 and His177), which interact with the 2-OH (Fig. 4d). Alanine mutations of the phosphate clamp residues had a severe consequence on both activity (>92% loss) and G3P $K_m$ (5- to 36-fold increase) (Table 2). Mutants S35C and S35T show 4% and 17% of the activity compared with the wild type, suggesting that the phosphate clamping is not only hydroxyl selective but also sensitive to the hydroxyl position. Stringent too is the requirement of Arg45, which, when mutated to lysine, caused over 90% loss of activity, and a 14-fold increase of G3P $K_m$. Similarly, mutating Lys104 to arginine reduced the activity to 25%, although with little effect on G3P $K_m$. For Asn180, adding one more carbon to the side chain (N180Q) impaired catalytic activity more than the N180A mutation (Table 2), suggesting the G3P-binding pocket is sensitive to subtle changes. Mutating Asn180 to histidine almost abolished activity, probably also for steric reasons. In addition, replacing Asn180 with the spatially similar aspartate caused over 95% loss of activity with a nearly 22-fold increase of $K_m$; presumably, charge–charge repulsion between the acidic residue and the phosphate of G3P impaired the substrate binding. For the binding of the 2-OH group, removing the hydroxyl of Ser142 caused 70% loss of activity. Remarkably, His177 was found to be indispensable with various side chains (Table 2 and Fig. 4e).

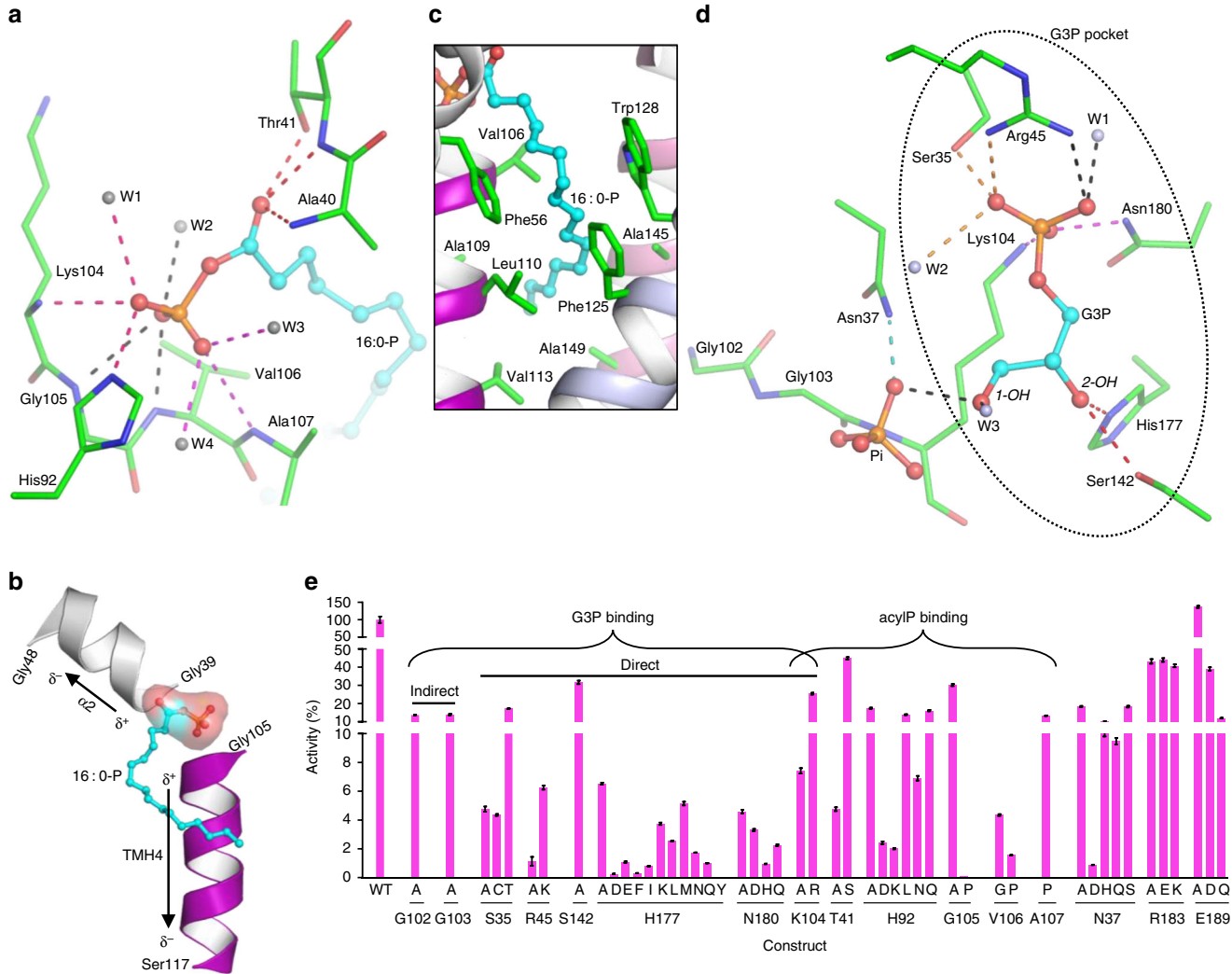

**Fig. 4** Interactions between *aa*PlsY and its substrates 16 : 0-P and G3P, and structure-based mutagenesis. **a** Interactions between the acylP head group and the enzyme. **b** The involvement of two α-helices in acylP binding. The aggregate effect of individual backbone microdipoles aligned along the α-helices axis (black arrows) causes a dipole moment with its positive pole at the N terminus and its negative pole at the C terminus. The resultant partial positive charges ($\delta^+$) at the N termini of the α2 ($n = 10$) and the TMH4 ($n = 13$) neutralize the phosphate charges of 16 : 0-P. **c** The hydrophobic groove that hosts the acyl chain of 16 : 0-P. Residues within 4.0 Å of the palmitoyl chain are shown as green sticks. **d** The G3P binding pocket. Dashed lines in **a** and **d** indicate distances within 3.2 Å. When appropriate, the dashed lines are colored differently for better visualization. **e** The activity of the wild-type and mutant enzymes. The wild-type activity ($34.55 \pm 3.14$ μmol min$^{-1}$ mg$^{-1}$, mean ± SD) was obtained from six independent experiments, and the mean value was set to 100%. Purified mutants were assayed under eleven different G3P concentrations without technical replicates. G3P concentrations typically span three orders of magnitude with the high concentrations saturating the system. The $V_{max}$ and SE from the Michaelis–Menten fitting are reported. Residues involved in binding of the two substrates are indicated appropriately

These extensive mutagenesis data show that the G3P-binding pocket is highly invariable. Consistent with this, analysis of 7,288 unique PlsY homolog sequences revealed that these residues are highly conserved (Table 2), suggesting the pin-point precision in steering G3P 1-OH for catalysis can only be achieved with the evolutionarily optimized combination (Supplementary Fig. 6).

**Rectification of functional motifs.** Previous biochemical studies with *sp*PlsY identified three functionally important motifs: Motif 1 contains residues 34–45 (*aa*PlsY numbering by sequence alignment) for acylP binding, Motif 2 contains residues 100–107 for G3P binding, and Motif 3 contains residues 177–189 for catalysis[8,12]. However, the substrate- and product-bound structures revealed that the both Motif 1 and Motif 3 are mainly responsible for G3P binding, whereas Motif 2 is mostly involved

in acylP binding (Fig. 4). The cause of discrepancies will be discussed.

**Pathways for substrate access and product egress.** The structures also reveal a plausible access/egress path for the lipid substrate/product to enter/leave the active site. Superimposing all MAG molecules near the active site of all five structures on the *aa*PlsY$_{16:0-P}$ structure reveals a gate-like structure between TMH2 and TMH5 (Fig. 5a, b). As noted, MAG resembles acylP and lysoPA; and indeed, the MAG molecule in the active site is almost superimposable with the lipid substrate and product (Fig. 3b). Thus, the collective poses of the MAG molecules near the gate (Fig. 5a, b) are reminiscent of the snapshots for the exchange of acylP and lysoPA between the active site and the membrane. Similar lateral diffusion mechanisms for lipid

**Table 2 Activity of mutants and conservation of functionally important residues**

| Construct | Activity[a], % | G3P $K_m$ (mM), (number-fold increase)[b] | $k_{cat}$ (s[−1]) | $k_{cat}$ / $K_m$ (s[−1] mM[−1]), (number-fold decrease)[c] | Conservation, % |
|---|---|---|---|---|---|
| WT | 100 | 1.39, (1.0) | 13.4 | 9.6, (1.0) | |
| S35A | 4.76 | 13.75, (9.9) | 0.64 | 0.05, (205.8) | 92.5, D (2.7)[d], T (0.9) |
| S35C | 4.35 | 20.47, (14.7) | 0.59 | 0.03, (334.9) | |
| S35T | 17.25 | 5.01, (3.6) | 2.33 | 0.47, (20.7) | |
| N37A | 18.38 | 3.04, (2.2) | 2.49 | 0.82, (11.8) | 88.3, S (6.0), G (2.0) |
| N37D | 0.88 | 13.72, (9.8) | 0.12 | 0.01, (1,109.5) | |
| N37H | 10.10 | 10.07, (7.2) | 1.37 | 0.14, (71.0) | |
| N37Q | 9.48 | 14.24, (10.2) | 1.28 | 0.09, (107.0) | |
| N37S | 18.31 | 1.25, (0.9) | 2.48 | 1.99, (4.9) | |
| T41A | 4.75 | 0.43, (0.3) | 0.64 | 1.49, (6.5) | 86.4, S (3.1), R (3.0) |
| T41S | 44.86 | 1.98, (1.4) | 6.07 | 3.07, (3.1) | |
| R45A | 1.13 | 50.46, (36.2) | 0.15 | 0.003, (3,189.8) | 92.2, A (1.4), K (1.0) |
| R45K | 6.25 | 19.32, (13.9) | 0.85 | 0.04, (220.2) | |
| H92A | 17.48 | 2.64, (1.9) | 2.37 | 0.9, (10.8) | 97.8, N (0.5), Q (0.1) |
| H92D | 2.42 | 6.31, (4.5) | 0.33 | 0.05, (185.7) | |
| H92E | NA[e] | | | | |
| H92K | 2.02 | 44.61, (32.0) | 0.27 | 0.01, (1,576.5) | |
| H92L | 13.91 | 5.43, (3.9) | 1.88 | 0.35, (27.8) | |
| H92N | 6.91 | 1.26, (0.9) | 0.93 | 0.74, (13.0) | |
| H92Q | 16.15 | 5.56, (4.0) | 2.19 | 0.39, (24.5) | |
| G102A | 13.74 | 9.97, (7.2); 7-fold increase[f] | 1.86 | 0.19, (51.7) | 97.9, S (0.5), R (0.1) |
| G103A | 13.97 | 6.74, (4.8); 7-fold increase[f] | 1.89 | 0.28, (34.3) | |
| K104A | 7.42 | 6.58, (4.7) | 1.00 | 0.15, (63.2) | 94.8, R (2.6), G (0.5) |
| K104R | 25.47 | 2.00, (1.4) | 3.45 | 1.72, (5.6) | |
| G105A | 30.20 | 3.58, (2.6) | 4.09 | 1.14, (8.5) | 73.2, A (21.3), S (3.4) |
| G105P | 0.09 | 2.08, (1.5) | 0.01 | 0.01, (1,698.9) | |
| V106G | 4.35 | 0.98, (0.7) | 0.59 | 0.60, (16.1) | 79.2, I (13.6), A (2.0) |
| V106P | 1.56 | 21.84, (15.7) | 0.21 | 0.01, (995.1) | |
| A107P | 13.32 | 21.74, (15.6) | 1.80 | 0.08, (116.3) | 88.1, S (6.2), L (1.7) |
| S142A | 31.69 | 2.89, (2.1) | 4.29 | 1.48, (6.5) | 52.6, A (33.0), G (5.2) |
| H177A | 6.52 | 9.24, (6.6) | 0.88 | 0.10, (100.9) | 91.4, N (2.2), F (1.0) |
| H177D | 0.25 | 82.13, (59.0) | 0.03 | 0.0004, (23,028.0) | |
| H177E | 1.08 | 41.08, (29.5) | 0.15 | 0.004, (2,709.5) | |
| H177F | 0.32 | 33.27, (23.9) | 0.04 | 0.0013, (7,493.0) | |
| H177I | 0.79 | 15.34, (11.0) | 0.11 | 0.01, (1,381.2) | |
| H177K | 3.74 | 13.63, (9.8) | 0.51 | 0.04, (259.6) | |
| H177L | 2.54 | 3.78, (2.7) | 0.34 | 0.09, (105.9) | |
| H177M | 5.14 | 3.20, (2.3) | 0.70 | 0.22, (44.3) | |
| H177N | 1.74 | 10.50, (7.5) | 0.24 | 0.02, (429.4) | |
| H177Q | 1.00 | 3.08, (2.2) | 0.14 | 0.04, (218.9) | |
| H177Y | 0.03 | 0.94, (0.7) | 0.004 | 0.005, (2,055.5) | |
| N180A | 4.59 | 26.31, (18.9) | 0.62 | 0.02, (408.7) | 89.1, D (1.7), G (0.5) |
| N180D | 3.31 | 30.37, (21.8) | 0.45 | 0.01, (652.7) | |
| N180H | 0.95 | 19.95, (14.3) | 0.13 | 0.01, (1,490.5) | |
| N180Q | 2.26 | 23.37, (16.8) | 0.31 | 0.01, (737.3) | |
| R183A | 43.17 | 3.84, (2.8) | 5.84 | 1.52, (6.3) | 73.4, K (7.1), N (5.3) |
| R183E | 43.94 | 10.33, (7.4) | 5.95 | 0.58, (16.7) | |
| R183K | 40.77 | 2.37, (1.7) | 5.52 | 2.33, (4.1) | |
| E189A | 136.94 | 5.79, (4.2) | 18.53 | 3.20, (3.0) | 88.0, G (1.1), K (0.5) |
| E189D | 39.03 | 2.81, (2.0) | 5.28 | 1.88, (5.1) | |
| E189Q | 12.01 | 15.10, (10.8) | 1.63 | 0.11, (89.6) | |

[a]Activities are shown as percentages to that of the wild-type (34.55 µmol min[−1] mg[−1]). See Fig. 4e for activity measurement of mutants
[b]Increased folds compared with the G3P $K_m$ of the wild-type $aa$PlsY
[c]Decreased folds compared with the specificity constant of the wild-type
[d]Numbers in brackets indicate frequency of the alternative residues found in 7,288 unique sequences of homologs from a BLAST[69] search against the InterPro 60.0 database[70]
[e]Purification of this mutant was unsuccessful. [f]Data taken from literature[12]

substrates/products have been proposed in the literature for membrane-integral enzymes such as the DgkA[27], the rhomboid proteases[28], the phosphatidate cytidylyltransferase[29], the phosphatidylinositol-phosphate synthase[30], the lipoprotein signal peptidase[31], the prolipoprotein diacylglycerol transferase[26], and the apolipoprotein N-acyltransferase[32,33].

The phosphate clamp for G3P binding is wide open to the cytoplasm, proposedly facilitating the G3P access to the active site. After the reaction, product Pi (from acylP) needs to be released from the phosphate hole to the cytoplasm. The Pi-bound structure $aa$PlsY$_{Pi}$ offers insights into this process. As shown in Fig. 5c, three phosphates are trapped in a positive patch stretching from the phosphate hole to the phosphate clamp, displaying a clear path for Pi release (Fig. 5c, d). Arg183 is located near the phosphate clamp (Fig. 5d) and probably help in extracting Pi by charge–charge attraction. In line with this, the R183A mutant was 43% active compared with the wild type (Table 2 and Fig. 4e). Conversely, the nearby Glu189 (Fig. 5d) might hinder the Pi

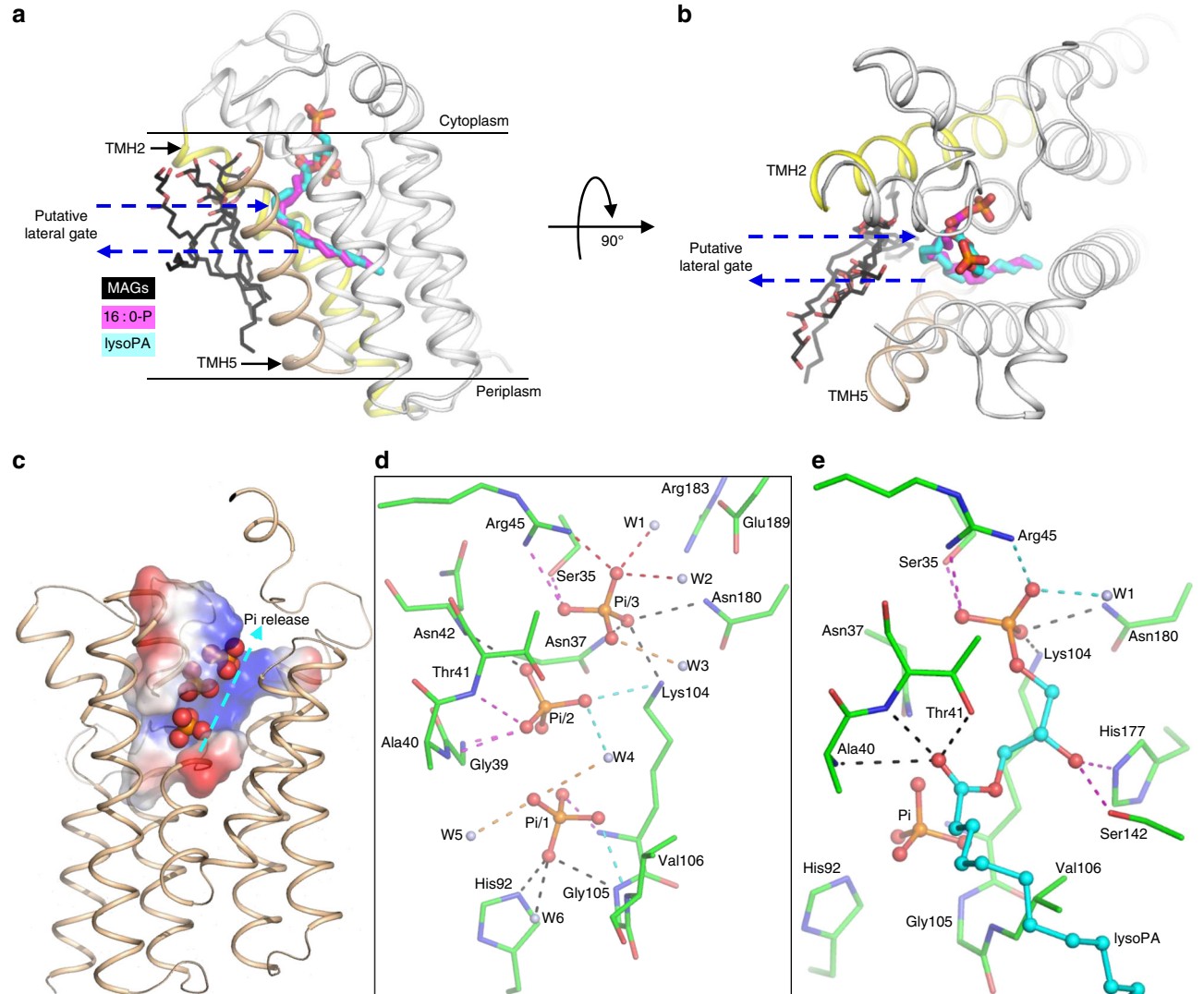

**Fig. 5** Structure insights into substrate access, product egress, and product inhibition. **a**, **b** The putative lateral gate for acylP access and lysoPA egress. **a** View from the membrane; **b** view from the cytoplasm. The carbon atoms of MAGs, 16 : 0-P and lysoPA are colored black, magenta, and cyan, respectively. The gate-forming TMH2 and TMH5 are colored yellow and wheat, respectively. **c** Overview of the *aa*PlsY structure with Pi bound. The region involved in phosphate binding are shown as electrical potential surfaces. Positively and negatively charged surfaces are shown as blue and red, respectively. **d** Interactions between the three phosphates and *aa*PlsY. **e** Interactions between lysoPA and *aa*PlsY. In **d** and **e**, Dashed lines indicate distances within 3.2 Å. When appropriate, the dashed lines are colored differently for better visualization

release by charge-charge repulsion. Eliminating the negative charge by alanine mutation caused enhanced activity (137% compared with the wild type; Table 2 and Fig. 4e). Interestingly, *aa*PlsY E189A was expressed normally, but its equivalent in *sp*PlsY (E197A) reportedly failed to assemble into the membrane[12]. It is possible that the two counterparts have different roles in the folding of the respective orthologs given the low homology in this region (Supplementary Fig. 1a).

Our work also suggested the order for the release of the two products. As shown in the *aa*PlsY_lysoPA structure, the phosphate group of lysoPA is located in the phosphate clamp (Fig. 5e), which is the same site for Pi/3 (Fig. 5d). Thus, the path for Pi release is blocked by lysoPA, suggesting that the lysoPA liberation is requisite for Pi release.

**Product inhibition.** LysoPA is a product inhibitor with IC_50 of 20 μM (ref. 12). Our structures show that lysoPA interacts with all G3P-binding residues (Figs. 4d and 5e), and forms three additional H-bonds to backbone amides of Ala40 and Thr41, and the

Thr41 hydroxyl. Furthermore, its hydrocarbon tail is anchored in the hydrophobic groove as seen for 16 : 0-P binding (Fig. 4b). Therefore, lysoPA should bind to PlsY with much higher affinity than does for G3P, featuring a competitive inhibitor. The release of the product inhibitor is probably driven by high concentrations of the substrates. In addition, the exiting Pi along the Pi-release path might aid the lysoPA clearance by electro-repulsion and competing for the phosphate clamp site.

**Catalytic mechanism.** It has been postulated that PlsY acylates G3P in a similar mechanism to the 'Asp-His' dyad thioester GPATs, based on the fact that PlsY contains a critical histidine residue[12] (His177 in *aa*PlsY). Thus, despite the absence of a paring Asp, His177 would act as a catalytic base to deprotonate G3P 1OH, similar to the case in PlsB[13] and the squash–spinach chimeric GPAT[34] (Fig. 6a). However, this mechanism cannot be invoked for PlsY with our data. First, the distance between the His177 NE2 and G3P 1-OH is 5.4 Å (Fig. 6b), which is not conducive for deprotonation; in fact, the hydrophobic

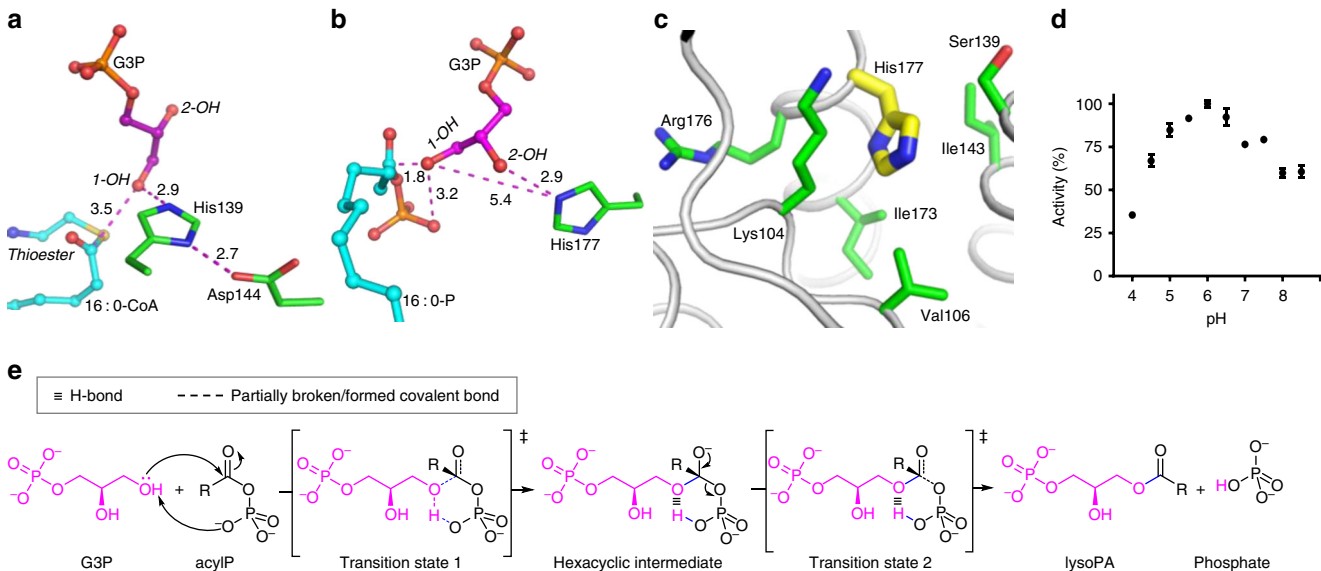

**Fig. 6** The catalytic mechanism of PlsY. **a** The spatial arrangement between the 'Asp-His' dyad and the docked substrates of the squash–spinach chimeric GPAT[34]. The critical His139 is close to the G3P 1-OH, suitable for deprotonation. **b** The spatial arrangement between His177 and the two substrates from *aa*PlsY$_{16\,:\,0-P}$ and *aa*PlsY$_{G3P}$. His177 is 5.4 Å apart with the G3P 1-OH, unsuitable for deprotonation. Dashed lines in **a** and **b** indicate distances (Å). **c** The hydrophobic environment of His177. **d** pH dependence of *aa*PlsY activity. Data represent the average ± standard deviation from triplicate measurements. **e** The 'substrate-assisted catalysis' mechanism

microenvironment of His177 (Fig. 6c) would prevent its protonation as the process will generate an unwelcome positive charge. Consistently, the calculated p$K_a$ of His177 is 4.42 (ref. 35), featuring a poor general base. Second, unlike the absolute requirement of His306 for PlsB activity[34], *aa*PlsY H177A remains considerably active (6.5% of that for the wild-type enzyme) (Fig. 4e), with an elevated G3P $K_m$ (6.6-fold increase). This indicates more of a substrate binding deficient (for both affinity and proper orientation) than a catalytic incompetent mutation. Third, the 'Asp-His-Ser' triad chymotrypsin, which shares essentially the same mechanism with the HX$_4$D GPATs, was reportedly 10% active at pH 6.0 (ref. 36), reflecting the strong pH dependence of the catalytic His57. By contrast, *aa*PlsY remained 35% active even at pH 4.0 (Fig. 6d), suggesting a different mechanism.

Unlike the thioesters, acylP can serve as a proton acceptor itself. Therefore, we propose a 'substrate-assisted mechanism'[37] for PlsY (Fig. 6e). G3P 1-OH is clamped in close proximity to the acylP phosphate (3.2 Å when superimposed; Supplementary Fig. 7a), facilitating its nucleophilic attack on the carbonyl carbon (1.8 Å when superimposed) (Fig. 6b, e). A chemically favorable hexacyclic transition state then forms, followed by a nucleophilic acyl substitution process through formation and collapse of the carbonyl tetrahedron (Fig. 6e). The oxyanion intermediate (Fig. 6e) is probably stabilized by the phosphate hole and Asn37. Consistent with this role, Asn37 interacts with neither substrates (Supplementary Fig. 5b); yet its alanine mutation caused 80% loss of activity, with no significant G3P $K_m$ defects. Moreover, substitution of Asn37 with the spatially similar aspartate almost abolished activity (Table 2); presumably, aspartate interferes with the intermediates by charge-charge repulsion. We performed quantum mechanics/molecular mechanics (QM/MM) modeling studies and observed key hexacyclic intermediates (Supplementary Fig. 7b-7e), providing additional supporting evidence for the mechanism.

## Discussion

In bacteria, the PlsX/PlsY/PlsC system is the most ubiquitous pathway for PA biosynthesis. The reaction of the water-soluble PlsX generates acylP, which is used by PlsY for lysoPA production. The membrane-bound PlsC then transfers the acyl group from acyl-CoA/ACP to lysoPA, generating PA. The crystal structures of PlsX (ref. 38) and PlsC[39] were reported in 2009 and during the submission of this manuscript, respectively. Our work on the bridging PlsY now completes the structure elucidation of the pathway.

For PlsX, the structure[38] and biochemical data[5] showed a homo-dimeric assembly. However, without substrates/products-bound structures or mutagenesis data, the active site architecture and catalytic mechanism remain unclear, although a hydrophobic groove at the dimer interface has been proposed as the active site[38]. For PlsC, the structure revealed a two-helix motif which anchors the protein to the membrane in such ways that the active site lies on the membrane leaflet and can be accessible by both the lipid substrate lysoPA and the more water-soluble substrate acyl-CoA/ACP[39]. Our PlsY structures show that the active site is half-way through the cytoplasmic side of the membrane and the catalysis happens in a hydrophilic interior deep in the lipid bilayer. The very different localization of the active sites prompt questions regarding the pathway: how are the intermediates (acylP and lysoPA) transferred from one enzyme to another? Do these enzymes act independently, or function as a higher order transient/stable complexes, so that the intermediates are directed to the enzymes without being released into the membrane?

AcylP is very unstable, prone to hydrolysis, and decomposes at temperatures above 60 °C (ref. 40). The origin of *aa*PlsY is the hyperthermophile *A. aeolicus* that grows between 85–95 °C, under which conditions acylP would rapidly decompose if left unprotected. Therefore, it would appear sensible, at least in the thermophiles, that acylP is escorted to or near PlsY's active site to avoid hydrolysis. Such 'substrate channeling' mechanisms for unstable intermediates have been reported[41]. As the acylP-producing enzyme, PlsX would be a strong candidate as the escorting protein.

Alternatively, the escort could involve another acylP-producing enzyme, the fatty acid kinase FakA/FakB[42,43]. In this system, FakA phosphorylates exogenous fatty acids bound to FakB1/B2, forming acylP. Upon finishing the reaction, FakB might dissociate

from FakA and transport acylP to PlsY. Similar escorting mechanisms for unstable lipid substrates has also been postulated; the 5-lipoxygenase may rely on the helper protein 5-LO activating protein for arachidonic acid delivery[44].

Fusion events could occur at some point along the evolution for genes encoding subunits of a complex[45]. We searched the Pfam database[46] and no hits were obtained for PlsX/PlsY or PlsY/PlsC. However, two entries (Uniprot A0A0P6X7I7 and A0A0P6XUK7) were found to have a DagV family protein fused to the C-terminus of PlsY. This is of particular interest since the DagV family members characterized so far are fatty acid binding proteins[42], including the aforementioned FakB1/B2. Indeed, the DagV part of the two entries shares 22.3–26.3% sequence identity and 44.7–47.1% similarity with FakB1/B2. Furthermore, the two fusion proteins are from the thermophilic bacteria, *Bellilinea caldifistulae* and *Thermanaerothrix daxensis*, which grow between 45–65 and 50–73 °C, with optimum temperatures of 55 and 65 °C, respectively[47]. These facts support the aforementioned hypothesis that substrate channeling between FakB and PlsY might have evolved to avoid acylP hydrolysis under high temperatures.

In addition, how PlsY was assayed in a previous report[12] provides supportive implications for substrate channeling. Thus, PlsY activity was also measured in membranes; however, unlike the current study where acylP was preloaded in the membrane, it was added as an aqueous solution[12], presumably in micelle form given its detergent-like chemical structure. It appears in that case, PlsY, which sits in the phospholipid bilayer, extracts acylP, which sits in the micelles (excepting those that had partitioned into membrane), to carry out the acylation. This mode of action would mimic the scenario whereby the acylP is handed over from water-soluble acylP-binding proteins to PlsY.

Having raised the possibility of substrate channeling, we would note that PlsY's activity does not depend on other proteins in vitro, and that our LCP assays and structure data suggest acylP in lipid bilayer can access PlsY's active site through a plausible gate that is open sidewise to the bulk membrane. Therefore, we propose that substrate channeling might exist, but it is not necessary for PlsY's function.

Although the PlsY's product lysoPA is very stable (unlike acylP), it is a potent membrane-disruptive detergent and signaling molecule. Thus, substrate channeling might also exists between PlsY and PlsC, in order to avoid lysoPA accumulation, which is potentially toxic to the cells. Further docking and experimental studies will help clarify this possibility.

One of the major goals of this study was to uncover whether PlsY uses a different catalytic mechanism. For such studies, it is desirable to have high-resolution structures of the enzyme-substrate/product complex(es). This is generally challenging due to the transit nature of the interactions between enzymes and substrates/products. For example, the water-soluble, squash-spinach chimeric GPAT, which is the only other GPAT with known structure, crystallographically has no substrates bound. In the current work, by extensive screening, we determined five high-resolution structures (Table 2) of *aa*PlsY, providing atomic details of the interactions between substrates/products and the enzyme. The structures reveal a completely different 'substrate-assisted catalysis' mechanism, which adds one more distinct feature to PlsY and expands the current knowledge regarding acylation biocatalysis.

Several discrepancies were found between the previous biochemistry-based model[12] and our crystal structures, regarding PlsY's topology (Fig. 2c) and functional motifs. First, PlsY adopts a 7-TMH membrane topology, as opposed to the previously reported 5-TMH model[12]. The latter was determined elegantly using the SCAM analysis, through which the localization of selected residues is assigned based on the Cys-accessibility to

water-soluble thio-reactant in responding to membrane integrity. Importantly, all SCAM assignments agree with the current 7-TMH topology, with the exception of Cys121 (Fig. 2c). This residue reacted with $N^\alpha$-(3-maleimidylpropionyl)-biocytin only when the membrane was disrupted by toluene, leading to the deduction that Cys121, as well as the adjacent regions, were located in the cytoplasm. The current model (Fig. 2a, c) rationalizes this interpretation and show that Cys121 on the TMH5 is buried in the membrane. This localization is consistent with the previous observation that Cys121 showing no accessibility to the water-soluble thio-labeling reagent when the membrane is intact. We further reasoned that the glycine-rich composition and the lack of long hydrophobic helices in the TMH4-5 region might have down-ranked them as TMHs by visual inspection and prediction algorithms. Second, the previous study suggested the Motif 2 containing residues 100–107 is responsible for G3P binding[12]. The main evidence was that the mutation of Gly102 and Gly103 to alanine both caused $K_m$ defects for G3P[12]. Although these biochemical data are replicated in the current study (Table 2), the structures show the majority of this loop is directly involved in acylP binding by contributing to the phosphate hole (Fig. 4a). Nevertheless, the structures rationalize the effect of Gly102 and Gly103 on the G3P binding. Thus, these two glycine residues, together with Gly105, facilitate the localization of the sandwiching Lys104 onto a sharp turn in Loop3 for the G3P binding (Fig. 4d). Replacing the structurally flexible glycines with alanines will deform the current geometry and misorient Lys104, thus indirectly affecting the G3P binding. Third, the Motif 3 (residues 177–189 in *aa*PlsY) was proposed to be responsible for catalysis[12], based on the experimental evidence that His185 in *sp*PlsY (the equivalent of His177 in *aa*PlsY) was sensitive to mutation. Agreeing with this fact, residues with varies chemical properties and spatial volumes were introduced at this position but none could replace His177 (Fig. 4d). Disagreeing with the previous assignment, however, the structure shows that His177's role is for G3P binding, not for catalysis. Thus, similar to Ser35/Arg45/Asn180, His177 is indispensable not because it is a catalytic base, but rather the acylation process requires precise G3P positioning through the interaction with these residues.

The crystals used for structure determination were grown in LCP. To demonstrate the functional relevance of these structures, it was important to show that PlsY was enzymatically active in LCP, a somewhat unusual membrane mimetic. Accordingly, a coupled assay was developed wherein the phosphate-releasing activity of LCP-reconstituted PlsY was quantified using a Pi-biosensor. Unlike hyperthermophile-origin enzymes that are inactive at low temperatures (20–37 °C)[48], *aa*PlsY was active at 30 °C, showing a maximal activity of 34.55 μmol min$^{-1}$ mg$^{-1}$ or $k_{cat}$ of 13.4 s$^{-1}$. The activity data demonstrated that, although *aa*PlsY is highly thermostable (Supplementary Fig. 2c), it still possesses conformational flexibility that is necessary to carry out the reaction. In addition, the catalytic efficiency is comparable to that of DgkA in LCP[16], supporting the view that the relatively inflexible active site (Fig. 3b, d) is a feature of the PlsY super-family, rather than an ultra-stable enzyme at low temperatures showing rigid, less-active conformations.

Compared with the kinetic parameters of *aa*PlsY (Fig. 1d, e), lower $V_{max}$ and $K_m$ values of 18 nmol min$^{-1}$ mg$^{-1}$, 30 μM 16 : 0-P, and 92 μM G3P, have been reported for partially purified *sp*PlsY[12]. Higher specific activity obtained in the current study is expected because pure enzyme was used. However, the 2,000-fold higher value would mean that, if the two orthologs have equal catalytic efficiency, the abundance of *sp*PlsY in the membrane from the previous recombinant *E. coli* strain[12] would be at 0.05%. This is lower than what would be expected, even when considering the potential toxicity by overexpressed membrane

proteins. Therefore, by rough estimation, it appears the *sp*PlsY is catalytically less competent than *aa*PlsY, although it must be appreciated that the two assays were conducted in different membrane environments. Further experiments with pure proteins will provide proper comparisons. The $K_m$ of 16 : 0-P from the two studies are also not directly comparable, because this substrate is confined in the lipid bilayer in the current study, whereas it was added as a solution previously[12]. The apparent $K_m$ of G3P for *aa*PlsY in LCP is 15-folds higher than reported[12]. Apart from the possible intrinsic G3P affinity differences between the two, slow diffusion of substrates in LCP would result in operational $K_m$ values that are higher than true $K_m$, as discussed in a previous study with DgkA[16]. Despite the different apparent $K_m$, we note that the data are consistent across the two studies regarding the extent of $K_m$ defects for the mutants G102A and G103A[12] (Table 2).

Compared with the existing PlsY activity assay that uses [14]C-labeled G3P[12], the LCP-based assay is continuous, does not use radioisotopes, and can be performed in a 96-well format for high-throughput screening. The latter is critically important for drug discovery applications. The method should prove generally applicable for a range of phosphate generating membrane enzymes. Moreover, the successful measurement of 16 : 0-P $K_m$ should encourage the application of LCP in kinetic assays that employ hydrophobic substrates with low solubility in detergents, such as cholesterol metabolising enzymes.

Being unique to bacteria and vital for phospholipid biosynthesis, PlsY is an attractive antibiotic drug target[5,9,10]. As revealed by the sequence homology analysis (Table 2 and Supplementary Fig. 6), the active site residues between different species are highly conserved. Thus, our *aa*PlsY structure should facilitate homology modeling of pathogen-origin PlsY for virtual inhibitor screening. And our high throughput assay should help characterize, design, and improve the small molecules, ultimately contributing to the development of novel therapeutics to combat multi-drug resistance.

## Methods

**Expression and purification of PlsY.** The *plsY* genes from *A. aeolicusi*, *S. pneumoniae*, *Thermotoga maritime*, and *Thermus thermophiles* were codon-optimized and synthesized by PCR using overlapping primers. The *plsY* gene from *E. coli* was PCR-amplified from genomic DNA. Genes were cloned into a pET-backboned vector with a C-terminal enhanced green fluorescent protein (eGFP) tag termed p3EG. Mutants were made using PCR-based site-directed mutagenesis. Sequence information regarding constructs and primers (for cloning and mutagenesis) are listed in Supplementary Table 1.

*E. coli* BL21 (DE3) cells (catalog number C600003, Thermo Fisher) carrying plasmid p3EG-*aa*PlsY were grown at 20 °C in M9 medium (40 mM Na₂HPO₄, 20 mM KH₂PO₄, 10 mM NaCl, 20 mM NH₄Cl, 0.1 mM CaCl₂, 1 mM MgSO₄, 0.4% (w/v) glucose, 50 mg L⁻¹ kanamycin, and 1/8 tablet per liter of a Wyeth Centrum Multivitamins) to OD₆₀₀ of 0.6–0.8 and were induced with 0.05 mM IPTG for 20–22 h.

For seleno-methionine (SeMet) labeling, cells were grown to OD₆₀₀ of 0.6 as for the native protein. Before induction, methionine biosynthesis was supressed[49] by adding 100 mg L⁻¹ of Lys, Phe, and Thr, and 50 mg L⁻¹ of Ile, Leu, and Val. After 30 min, L-SeMet was added to 60 mg L⁻¹, and IPTG was added to 0.05 mM for induction at 20 °C for 22 h.

Native and SeMet protein purification were performed at 4 °C unless specified otherwise. Biomass (75–85 g) re-suspended in 260 mL Buffer A (150 mM NaCl, 0.1 mM EDTA, 1 mM phenylmethylsulfonyl fluoride (PMSF), 8 mM MgCl₂, 10 µg mL⁻¹ DNase I, 50 mM Tris-HCl pH 8.0) was passed through a cell disruptor at 25 kpsi for three times. The cell lysate was centrifuged at 20,000 *g* for 30 min. The supernatant was centrifuged at 150,000 *g* for 1.5 h to yield membranes, which were re-suspended with 100–200 mL of Buffer B (150 mM NaCl, 10% (v/v) glycerol, 0.1 mM EDTA, 1 mM PMSF, 50 mM Tris-HCl pH 8.0), and solubilized with 1% (w/v) of *n*-dodecyl-β-D-maltoside (DDM, Prizen Biomedical, Inc., Shanghai, China). After 1.5 h, insoluble materials were removed by centrifugation at 48,000 *g* for 1 h. The supernatant was heated at 65 °C for 10 min, centrifuged at 20,000 *g* for 30 min, and incubated with 30–35 mL of Ni-NTA resin. After 1.5 h of batch binding, the resin was washed sequentially with 300 mL of 150 mM NaCl in Buffer C (0.03% (w/v) DDM (catalog number D310S, Anatrace, Maumee, OH, USA), 50 mM Tris HCl pH 8.0], 300 mL of 1 M NaCl, 10 mM imidazole in Buffer C, and

450 mL of 150 mM NaCl and 45 mM imidazole in Buffer C. The fusion protein was eluted using 0.25 M imidazole and 150 mM NaCl in Buffer C, desalted and digested with 3C protease at molar ratio of 1 : 6 (protease : *aa*PlsY-GFP) in the presence of 0.1 M Na₂SO₄ for 16 h at room temperature (RT). The mixture was loaded onto a fresh Ni-NTA column to remove GFP and 3C protease. The flowthrough was concentrated in a 50-kDa cut-off concentrator to 6–8 mg mL⁻¹, and loaded onto a Superdex 200 10/300 GL column pre-equilibrated with Buffer D [0.03% (w/v) DDM, 150 mM NaCl, 20 mM Tris-HCl pH 8.0]. The elution was concentrated to 20 mg/mL for crystallization.

For enzymatic assays, the wild-type and mutant *aa*PlsY were expressed as above but replacing the His-eGFP tag with His₈ tag (termed p3EC). The expression conditions and purification procedures followed that for the GFP fusion protein but omitting the 3C protease digestion and gel filtration steps.

**Production of the 3C protease.** A DNA fragment containing the coding sequence, from 5′–3′, for GST, 3C protease site (LEVLFQGP), octaHis tag, and the 3C protease were cloned into a pET vector. *E. coli* BL21 (DE3) cells carrying this plasmid was induced with 1 mM IPTG for 20 h at 20 °C. Cell pellets from 6 l of culture were resuspended in 210 mL Buffer E (5 mM MgCl₂, 150 mM NaCl, 1 mM PMSF, 1 mM tris(2-carboxyethyl)phosphine (TCEP), 10 µg mL⁻¹ DNase I, 20 mM imidazole, 20 mM Tris HCl pH 8.0) and lysed using a cell disruptor. After centrifugation at 20,000 *g* for 15 min, the supernatant was incubated with 4 mL of Ni-NTA resin and 0.5 mM PMSF for 1.5 h at 4 °C. The resin was then washed with 240 mL of 1 M NaCl in Buffer F (0.2 mM TCEP, 20 mM Tris HCl pH 8.0) and 400 mL of 150 mM NaCl, 50 mM imidazole in Buffer F. The protease was eluted 250 mM imidazole in Buffer F, desalted, concentrated to 4 mg mL⁻¹, flash frozen in liquid nitrogen, and stored at − 80 °C.

**Purification of the phosphate binding protein.** The coding sequence of residue Glu26-Tyr346 of the phosphate-binding protein (PBP) was amplified from the *E. coli* BL21 (DE3) genome and cloned into vector p3EC. The mutant A197C (ref. 14) was made using site-directed mutagenesis. *E. coli* BL21 (DE3) cells carrying the plasmid were induced with 1 mM IPTG at OD₆₀₀ of 0.6-0.8. Cells were lysed in Buffer G (0.5 mM TECP, 1 mM PMSF, 8 mM MgCl₂, 10 µg mL⁻¹ DNase I, 20 mM Tris-HCl pH 8.0) using a cell disruptor. Cell lysate removed of debris by centrifugation was incubated with 10 mL of Ni-NTA resin for 2 h. The resin was then washed with 1 liter of Buffer H (10 mM Tris-HCl pH 8.0) and 150 mL of 40 mM imidazole in Buffer H. Protein was then eluted using 250 mM of imidazole in Buffer H, desalted, and loaded onto a 5 mL HiTrap Q Sepharose FF column equilibrated with Buffer H. PBP was eluted using 40 mL of 0–100 mM NaCl in Buffer H at a flow rate of 1 mL min⁻¹, concentrated to 32.5 mg mL⁻¹, flash frozen in liquid nitrogen, and stored at − 80 °C.

**Fluorescence labeling of PBP.** Labeling PBP A197C with the thio-reactive *N*-[2-(1-maleimidyl)ethyl]-7-(diethylamino)coumarin-3-carboxamide (MDCC)[14] was performed as below. Contaminant Pi was removed in a 4 mL reaction containing 13 mg of PBP, 0.05 mg mL⁻¹ of purine nucleoside phosphorylase (PNPase), 0.2 mM of 7-methylguanosine (MEG) and 10 mM Tris-HCl pH 8.0 for 30 min at RT. MDCC was added to the mixture to 0.15 mM for labeling at RT for 2 h. Unreacted MDCC was removed with a Superdex 200 10/300 GL column pre-equilibrated with 10 mM Tris HCl pH 8.0. Labeled and unlabeled PBP was separated on a 5 mL HiTrap Q Sepharose FF column by 200 mL of 0–250 mM NaCl gradient elution at a flow rate of 1 mL min⁻¹. The concentration of MDCC-PBP was calculated by correcting the A₂₈₀ from the MDCC, using the formula[14]: [(A₂₈₀, ₁ cm − A₄₃₀, ₁ cm × 0.164) / 61,656 M⁻¹]. The MDCC-PBP was concentrated to 3.7 mg mL⁻¹, flash frozen in liquid nitrogen, and stored at − 80 °C.

**Purification of the PNPase DeoD.** The *deoD* gene from *E. coli* BL21 (DE3) was cloned into a pET-backboned vector with an N-terminal His₈ tag termed p3EN. *E. coli* BL21 (DE3) carrying the plasmid was induced with 1 mM IPTG for 6 h at 37 °C. Cells re-suspended in Buffer I (2 mM EDTA, 10% (v/v) glycerol, 0.5 mM TCEP, 50 mM NaCl, 1 mM PMSF, 8 mM MgCl₂, 10 µg mL⁻¹ DNAse I, 100 mM Tris HCl, pH 7.5) were lysed using a cell disruptor. The lysate was centrifuged at 140,000 *g* for 50 min at 4 °C. The supernatant was mixed with 10 mL of Ni-NTA resin. After 1 h of batch binding, the resin was washed with 300 mL of Buffer J (2 mM EDTA, 10% (v/v) glycerol, 100 mM Tris HCl pH 7.5), and 500 mL of 35 mM imidazole in Buffer J, and eluted with 0.25 M imidazole in Buffer J. The elution was desalted and loaded onto a 5 mL HiTrap Q Sepharose FF anion exchange column. Pure DeoD protein was obtained by 100 mL of 0–500 mM NaCl gradient elution in Buffer J at a flow rate of 1 mL min⁻¹. Protein was concentrated to 3.8 mg mL⁻¹, flash frozen in liquid nitrogen, and stored at − 80 °C.

**Measurement of detergent concentration.** Detergent concentration was measured using the phenol-sulfuric acid method[50]. Briefly, 6 µL of DDM solution ranging from 0.02–0.1% (w/v) was added to 90 µL of concentrate H₂SO₄. After addition of 18 µL of 5% (w/v) phenol, the mixture was heated at 90 °C for 5 min and cooled down to RT. A₄₉₀ was measured using a plate reader with 90 µL of solution in a 384-well plate. The protein samples were diluted 125 times before assayed. Measurements were performed in triplicates.

**Detergent removal using β-cyclodextrin.** A typical preparation of aaPlsY contains 4.5–5.0 %(w/v) DDM at a protein concentration of 20 mg mL$^{-1}$. The high concentration of DDM causes lamellar phase problem during crystallization when the detergent-like 16:0-P is included and the 7.8 MAG was used as the host lipid. Therefore, excess DDM was removed using β-cyclodextrin[51]. Pooled fraction from gel filtration (4.2 mL solution containing 0.77 mg mL$^{-1}$ aaPlsY and 0.5 %(w/v) DDM) was concentrated using a 50 kDa cut-off membrane to 2.1 mL before added with 2.4 mL of 10 mM β-cyclodextrin. The mixture was concentrated to 0.9 mL and added with 1 mL of 10 mM β-cyclodextrin. The diluted sample was again concentrated to 0.38 mL, added with 0.43 mL of β-cyclodextrin, and re-concentrated to 22 mg mL$^{-1}$. The protein solution (0.13 mL) was diluted 10-folds with buffer free of β-cyclodextrin or DDM, and concentrated to 0.13 mL again. The process was repeated once, to remove β-cyclodextrin, which would interfere with the downstream DDM concentration measurement[50] because of its sugar-containing moiety. Using this approach, DDM concentration was reduced from 4.5–5% (w/v) to 1.6–1.8% (w/v).

**Synthesis of palmitoyl phosphate.** Palmitoyl phosphate was synthesized as described below[5,52]. Briefly, 3.76 g Ag$_3$PO$_4$ and 2.08 g of 92% (w/v) H$_3$PO$_4$ were mixed and ground in a mortar on ice with 40 mL of cold ether. The suspension was stirred on ice for 1.5–2 h. Palmitic chloride (5 mL) dissolved in 20 mL of cold ether was added drop-wise to the mixture. The suspension was ground, stirred on ice for 20 min, before centrifuged at 4,000 g for 10 min at 4 °C. The supernatant was decanted to a round-bottom flask for solvents removal using a rotary evaporator at RT. Warm benzene (60 °C, 100 mL) was added to the dry compound, resulting in a turbid suspension, which was filtered before crystallization at 6–8 °C for 4 h. Crystals were collected by filtration, washed with cold benzene, and dried in vacuo (0.07 mBar, − 102 °C). Nuclear magnetic resonace (NMR) and mass-spec analysis confirmed the product, as previous reported[5]: $^{1}$H NMR (dimethyl sulfoxide (DMSO), 400 MHz), $\delta$ 0.85 (t, 3 H, J = 6.4 Hz), 1.24 (s, 26 H), 1.45–1.56 (m, 2 H), 2.38 (t, 2 H J = 7.2 Hz); Mass-Spec (electrospray ionization) $m/z$ = 335 [M 2 H]$^-$. The final product was protected from moisture and stored at − 80 °C.

**PlsY activity assay using Pi-biosensor.** The glycerol phosphate (1 M, catalog number G6501, Sigma) contains trace amount of Pi, the level of which was quantified as 0.1 mol% using the Pi-biosensor assay. As PBP is included at 7–10 μM for the LCP-based PlsY assay (see below), the contaminant Pi in G3P will saturate PBP in the Assay Mix with 10 mM G3P. Thus, it was necessary to remove the Pi contaminant. This was done using a Pi-scavenging system, through which the free Pi is transferred to MEG by DeoD, a reaction that can be monitored by the fluorescence drop of Pi-bound MDCC-PBP[14]. In detail, 1 M of G3P was incubated with MEG (5 mM) and DeoD (0.05 mg mL$^{-1}$) for 1 h at RT. The mixture was then heated at 60 °C for 10 min to inactivate DeoD, and the DeoD precipitates were removed by centrifugation at 20,000 g for 20 min at RT. The heat inactivation of DeoD was separately confirmed by the phosphorylase activity assay using MDCC-PBP.

LCP was made by mixing monoolein and aaPlsY solution at 1.5:1 volume ratio in a coupled syringe device[53]. The 16:0-P was added either as powder, or as 200 mg mL$^{-1}$ solution in DMSO, before mixing. To a half-area micro-plate well, 1 μL LCP was dispensed as described[16]. Residual Pi was removed by soaking the LCP with 160 μL of Pi-free buffer. Assay Mix (50 μL of various concentrations of glycerol phosphate, 7 μM MDCC-PBP, 150 mM NaCl, 50 mM Tris HCl pH 8.0) pre-warmed at 30 °C was added to the wells. Fluorescence (Ex.445 / Em.500 nm) was monitored at 30 s intervals with 3 s shaking between each read.

For G3P $K_m$ assay, 0–50 mM of glycerol phosphate, 1.8 mol% of 16:0-P (with respect to monoolein), and 1 μg mL$^{-1}$ of aaPlsY were used. For 16:0-P $K_m$ assay, 0–2 mol% of 16:0-P, 30 mM of glycerol phosphate, and 1.18 μg mL$^{-1}$ of aaPlsY were used.

For enzyme loading, 1.8 mol% of 16:0-P and 30 mM of glycerol phosphate were used. Two LCP samples, one with no enzyme and the other with 72 μg mL$^{-1}$ enzyme, were mixed at various ratios to achieve the desired loading.

For aaPlsY assay at various pHs, 3.2 μg mL$^{-1}$ of the enzyme, 20 mM of the glycerol phosphate, and 1.4 mol% the 16:0-P were used. The buffers in the Assay Mix were the following: sodium acetate/acetic acid for pH 4.0, 4.5, and 5.0; MES/Na for pH 5.5, 6.0, and 6.5. HEPES/Na for pH 7.0; and Tris HCl for pH 7.4, 8.0, and 8.5.

To accurately determine the amount of 16:0-P in LCP, we used a phosphorus assay[50]. LCP containing 16:0-P was soaked to free Pi and dried with at 55 °C for 30 min. Wet-ashing was performed by incubation of perchloric acid (130 μL) at 180 °C for 50 min. After cooling to RT, Milli-Q water (660 μL), 2.5% (w/v) ammonium molybdate (100 μL), and 10% (w/v) L-ascorbic acid (100 μL) were sequentially added to the sample. The mixture was heated at 100 °C for 5 min for colour development. A$_{810}$ of the sample and NaH$_2$PO$_4$ standards (0–1 μg phosphorus) treated as above were measured for phosphate content calculation.

Activity assays for $K_m$ determination (Fig. 2d, e), enzyme loading (Fig. 2f), and pH optimum (Fig. 6d) were carried out in triplicates from single experiments. Values reported are the average ± SD from the triplicate measurements.

For activity assay of mutants, the protein concentrations were adjusted to as high as 2.8 mg mL$^{-1}$ in order to increase measurement sensitivity. The concentrations of 16:0-P were in the range of 3.3–5.2 mol%. G3P $K_m$ values were determined for all mutants by measuring enzymatic activity under 11 different G3P concentrations with no technical replicates. The typical concentrations range from 0.2 to 200 mM. For some mutants where the activity does not reach saturation at 200 mM, the G3P concentrations were adjusted to 2.2–800 mM. No replicates were carried for the mutants. $V_{max}$ and $K_m$ values were determined using the 11 data points, in addition to the blank (0 mM G3P). The $V_{max}$ and SE from the Michaelis–Menten fitting were reported for all the mutants (Fig. 4e and Table 2).

As an indication of the reproducibility, the activity of the wild-type aaPlsY under standard conditions was at 34.55 ± 3.14 μmol min$^{-1}$ mg$^{-1}$ from six independent repeats. These repeats were performed periodically during the course of this study to check the integrity of reagents.

The enzyme concentration refers aaPlsY concentration in DDM before making LCP.

Enzymatic activity was calculated using $\Delta_{fluorescence\ intensity}$ / $\Delta_{time}$ in the linear range of the progress curve. Fluorescence intensity was converted to pmol of Pi according to calibrations using NaH$_2$PO$_4$ standards and MDCC-PBP, under various conditions specified for aaPlsY. $K_m$ and $V_{max}$ were calculated using the Prism 5.0 software.

**PlsY activity assay using TLC.** The product lysoPA was assayed using TLC. LCP loaded with 2 mol% 16:0-P and 1 mg mL$^{-1}$ of aaPlsY (in 0.5% DDM) was incubated with 20 mM glycerol phosphate in the Assay Mix for 20 min. Control experiments were carried out by omitting the enzyme. After the reaction, LCP was dried in vacuo (0.07 mBar, − 102 °C), followed by extraction with 480 μL of chloroform : methanol : acetic acid (19 : 1 : 1, v/v). Four microliters of extraction were spotted onto a TLC plate that had been pre-run in chloroform. The plate was developed in chloroform : methanol : acetone : acetic acid : water : hydrochloride (13 : 2 : 4 : 8 : 0.9 : 0.1, v/v), stained with 10% (w/v) phosphomolybdic acid in ethanol, and heated at 150 °C for stain development.

**LCP crystallization.** Protein was mixed with either equal volume[17] (for 7.8 MAG) or 1.5-folds (for 9.9 MAG) of lipids to form LCP. Crystallization trials were set up by covering 50 nL of LCP bolus with 800 nL of precipitant solution in 96-well glass sandwich plates using a Gryphon robot (Art Robbins, Sunnyvale, CA, USA)[54].

Crystals of aaPlsY$_{MAG}$ and the SeMet protein were grown at 20 °C in the 7.8 MAG LCP with 25–30% (v/v) of triethylene glycol, 0.1 M (NH$_4$)$_2$SO$_4$, and 0.1 M glycine/HCl pH 3.8 as precipitant solution. Crystals of PlsY$_{G3P}$ were grown at 20 °C in monoolein LCP, with 80 mM glycerol phosphate, 0.1–0.2 M LiCl, 35–38% (v/v) PEG 400, 0.1 M Tris HCl pH 8.0 as precipitant solution. Crystals of aaPlsY$_{lysoPA}$ were grown at 20 °C in monoolein LCP containing 1 mol% of lysoPA, with 39% (v/v) PEG 350 MME, 40 mM NaCl, 40 mM Tris HCl pH 8.0 as precipitant solution. Crystals of aaPlsY$_{16:0-P}$ were grown at 4 °C in 7.8 MAG LCP containing 1 mol% of 16:0-P, with 0.4–0.6 M KF, 22–26% (v/v) PEG 400, 0.1 M Tris HCl pH 8.0 as precipitant solution. It is noteworthy that the detergent concentration of the protein was lowered from 4.5% to 1.8% (w/v) for this condition. Crystals of aaPlsY$_{Pi}$ were grown at 4 °C in 7.8 MAG LCP, with 24% PEG 400 (v/v), 0.4 M Na$_2$HPO$_4$, 0.1 M HEPES pH 7.0 as precipitant solution.

**Data collection and processing.** Crystals were collected from the glass sandwich plates under microscope using MiTeGen Dual Thickness MicroMounts loops and directly cryo-cooled by plunging the crystals into liquid nitrogen without additional cryo-protectant[55,56]. X-ray diffraction data were collected at the BL18U1 or BL19U1 beamlines in the National Center for Protein Sciences Shanghai and BL17U1 at Shanghai Synchrotron Radiation Facility[57]. Diffraction data were processed with X-ray Detector Software (XDS)[58] and scaled and merged with Aimless[59].

**Structure solution and refinement.** Experimental phases were obtained from single-wavelength anomalous dispersion data[60] collected using SeMet-crystals. Three Se sites were found using SHELX[61]. An initial model with 187 residues was built by Phenix.Autobuild[62] with $R_{free/work}$ of 0.2715/0.2431. The model was further built manually using Coot[63] as guided by maps generated by Phenix.Refine[62], to contain residues 0–195 (the initial Met was numbered as 0). This model was later used as the search model for molecular replacement with Phaser to solve all other structures. Simulated annealing (SA) was performed in the initial stages of refinement to remove model bias. The interactive model building was done using Coot with gradually improved maps generated by Phenix.Refine.

For ligands, three types of electron density maps were calculated (Supplementary Fig. 4). Sigma-A weighted $2F_o$-$F_c$ SA omit map and $F_o$-$F_c$ SA omit map were obtained using models omitting ligands by Phenix.Refine[62] with SA. The composite omit map was calculated by systematically omitting the entire content of the unit cell using the phenix.composite_omit_map module[62].

**QM/MM modelling of the catalytic mechanism.** Modeling of the enzyme complexes was done using graphical user interface Maestro 10.4 in Schrödinger Suite 2015-4. The two substrates G3P and 16:0-P were taken from their respective structures aaPlsY$_{G3P}$ and aaPlsY$_{16:0-P}$ and put into the high-resolution structure aaPlsY$_{16:0-P}$. Five water molecules forming key interactions to the substrates were also included. The system was processed with the Protein Preparation Wizard

tool[64]. The intermediate was generated by modifying the bonding of the substrates. The aaPlsY substrates and aaPlsY-intermediate complexes were first optimized in OPLS3 force field, using the Refine Protein–Ligand Complex tool[65]. Then QM/MM approach in DFT-B3LYP method was run to find their minimums, using QSite 6.9 program[66]. Finally, the transition state for the addition step is obtained by QM/MM approach in the same method; the transition state searching method is linear synchronous transit from the aaPlsY substrates and aaPlsY-intermediate QM/MM minimums.

**Molecular docking**. Crystal structure of the squash-spinach chimeric GPAT (PDB entry 1IUQ) was processed using the same tool as in QM/MM modeling section. G3P was docked into the pocket with Glide 6.9 program[67]. The palmitoyl CoA was docked into the GPAT-G3P complex with the same docking program and optimized with the Prime (Schrödinger) program.

**Data availability**. Atomic coordinates and structure factors for the reported structures are deposited in the Protein Data Bank (PDB) under accession codes of 5XJ5 (aaPlsY$_{MAG}$), 5XJ6 (aaPlsY$_{G3P}$), 5XJ7 (aaPlsY$_{16:0-P}$), 5XJ8 (aaPlsY$_{lysoPA}$), and 5XJ9 (aaPlsY$_{Pi}$). All other relevant data are available from the corresponding author upon reasonable request.

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

## Acknowledgements

We thank Professor J. Chou at Harward Medical School and Professor Y. Huang at Biophysics Institute, Chinese Academy of Sciences (CAS), for their critical review of the manuscript, and Dr D. Yao for assistance and advice in X-ray data collection for the initial low-resolution SeMet data sets. D.L.'s lab is supported by the 1000 Young Talent Program, the Shanghai Pujiang Talent Program (15PJ1409400), the National Natural Science Foundation of China (31570748 and U1632127), the CAS-Shanghai Science Research Center (CAS-SSRC-YJ-2015-02), and Key Program of CAS Frontier Science (QYZDB-SSW-SMC037).

## Author contributions

Z.L. and Y.T. performed protein purification, crystallization, X-ray diffraction data collection, and enzyme kinetics assays. J.B. assisted with the molecular cloning and site-directed mutagenesis. Y.W. carried out docking, QM/MM simulation, and catalytic mechanism analysis with supervision from S.Z. Y.L. helped with formulating the catalytic mechanism analysis. D.L. advised protein purification and crystallization, collected diffraction data, solved the structures, and wrote the manuscript with inputs from Y.W. and S.Z.

## Additional information

**Competing interests:** The authors declare no competing financial interests.

