## [Peer Review File · Nature Communications]

Reviewers' comments:

Reviewer #1 (Remarks to the Author):

This paper reports the structure of PlsY. The data are comprehensive and convincing. The conclusions reveal a unique catalytic mechanism for an essential enzyme in Gram-positive pathogens. The results and conclusions are clearly presented. There are a couple of places the presentation could be improved, and the impact of the work would improve by discussion of the structural biology of the three step pathway that is conserved in pathogens. Is there any implications from the structures that suggest a deeper understanding of phospholipid synthesis as a process?

The authors should consider making comments about the recently published structure of PlsC (Nat. Struct. Biol. 24:666-671, 2017). This is the next enzyme in the PlsX/PlsY/PlsC pathway, and the PlsY structure completes the structural overview of the pathway. PlsX was determined previously. Now that there is a structural view of the entire pathway to phosphatidic acid in pathogens, is there any deeper understanding of how the pathway works as an integrated process that can be gleaned by examining the structures.

With respect to the pathway for acylIP to find the active site, do the authors think that acylIP must first be placed into the membrane and diffuse into the active site? AcylIP is produced by PlsX, and another way of asking the question is: Do the structures suggest release of acylIP into the membrane or is it possible that PlsX directly hands off its product. Likewise, is there any indication from the structures that PlsY and PlsC would form a complex, or do the authors interpret their structures indicate that the substrates diffuse out of one enzyme into the membrane, where they find the next step in the pathway. Exogenous fatty acids are activated to acylIP by their phosphorylation on a fatty acid binding protein. Could the acyl chain be handed off in this case, or is the acylIP first deposited into the membrane?

Not sure that calling PlsY an evolutionary orphan is the correct term. It may be a connection to the deep past when acylation reactions occurred before the emergence of acyl group carrier cofactors. But an orphan?

Fig. S2a. What was the solvent for the gel filtration chromatography. Could not find it in the methods. The text refers to this figure as a reconstitution of PlsY with LCP. Please clarify.

Reviewer #2 (Remarks to the Author):

The manuscript describes structural and functional studies of the enzyme glycerol 3-phosphate acyltransferase PlsY from *Aquifex aeolicus* (aaPlsY). A collection of five crystal structures are presented with different combinations of bound substrates and products, providing a framework for the complete description of substrate binding and catalysis in this enzyme class. Importantly, an enzymatic assay developed to enable monitoring of catalysis in lipidic cubic phase mixtures, also used for crystallization of the enzyme, supports the claim that the enzyme is active in this environment. The assay is used for extensive mutagenesis studies that support the claims made regarding substrate/product coordination and catalysis. Based on the crystal structures, the authors propose a novel enzymatic mechanism, that doesn't rely on an aspartate-histidine dyad involved in catalysis in other acyltransferase enzyme classes. The 'substrate-assisted catalysis' proposed involves a direct nucleophilic attack from the glycerol 3-phosphate hydroxyl to the carbonyl group of the acyl phosphate with proper coordination of the substrates through multiple interactions with enzyme residues. Finally, the authors provide a re-evaluation of previous findings in light of the crystal

structures (topology, assignment of residue functional roles), and offer explanations as to the reasons previous findings/conclusions have been reached.

Overall, this is a very comprehensive and impressive work that conclusively characterizes all functional aspects of this class of acyltransferase enzymes. The work will be of great interest to structural biologists and biochemists alike.

The manuscript is generally well-written and fairly easy to follow. The figures and tables are well-crafted (though see some suggestions below). The methods section provides enough detail for reproducing experiments independently.

Minor comments:

- 1) When comparing the fold of aaPlsY with other 7-TM proteins of known structure, the structure of bacterial diacylglyceryl transferase (Lgt) has been omitted from the comparison. Although the topology of Lgt is different compared to PlsY, the fold appears to be fairly similar based on visual inspection (PDB ID 5AZB). It is worth exploring the fold similarity and also commenting on any functional similarities since Lgt also accommodates lipidic substrates.
- 2) It is mentioned that TMH4 and $\alpha 2$ act as dipoles to neutralize the phosphate charge of acylIP. What are the characteristics of TMH4 and $\alpha 2$ that make them act as dipoles? Some clarifications on what makes them act as such would help with comprehension by the reader.
- 3) The number of independent measurements (n) to determine enzymatic activity of aaPlsY and associated mutants should be included in Fig. 4c and/or Table 2.
- 4) It is mentioned that lysoPA (the product of the reaction that also acts as an inhibitor) interacts with PlsY more extensively than G3P, and thus should have higher affinity for the enzyme than G3P. Then how is lysoPA cleared? Through competition with the substrate acylIP? In general, clearance of Pi is discussed but not clearance of lysoPA.

Suggestions for improving the manuscript:

1. p.3: "...deletion of both PlsY and the acylIP-synthesizing enzyme PlsX are not possible..." clarify (is it lethal?) and provide citation.
2. p.6 Fig. 1c: It is very difficult to discern the products/bands in the thin layer chromatography assay. The contrast of this panel must be improved.
3. p.6: In text a lag phase of "approximately 2min" is mentioned, but in Fig. 1 legend "The 3-min lag phase" is mentioned. Which of the two is true? Also in the corresponding panel (Fig. 1b) a discernible lag phase is not readily apparent and perhaps should be labeled for clarity.
4. p.6 Fig. 1f: Is there a reason the points were not fitted with a curve in the enzyme loading graph?
5. p.8 Fig. 2c: The color-coding of the topology diagram is overwhelming. At the very least a color-coding inset should be included. Also consider simplifying, perhaps dropping the color-matching to TM segments in (a) and (b), since the topology diagram is already organized per transmembrane segment.
6. p.10 Fig. 3a,b: These panels should be made somewhat bigger, as their small size requires

increased focus to follow the points being made.

7. p.31: "...to inactivate DeoD, which was experimentally confirmed." Add how inactivation was experimentally confirmed.

8. p.42 Sup. Fig. 2: Improve contrast of panel (b) (SDS-PAGE)

9. p.43 Sup. Fig. 3: In (b) a label for the third (from left) panel is missing (presumably Bax inhibitor motif protein based on the main text)

10. The manuscript is well written but does infrequently contain some grammatical/syntactical errors. A quick scan from a copy editor will help to eliminate them.

Reviewer #3 (Remarks to the Author):

Li et al. report a structural and functional analysis of the bacterial glycerol-3-phosphate acyltransferase (GPAT) PlsY, an integral membrane protein. This enzyme uses acyl-phosphate as an acyl donor rather than the commonly found substrates acyl-CoA or acyl-ACP. The structure and mechanism was previously unknown.

The authors now describe five high-resolution structures of a GPAT in multiple states and at high resolution, including substrate- and product-bound states. The enzyme does not seem to undergo large conformational changes during the reaction mechanism. The structures provide seminal and highly relevant insight the architecture of PlsY and provide important clues to its mechanism. Based on the structural insight and on extensive biochemical characterization, Li et al. propose a plausible and novel mechanism for the acylation reaction. The functional analysis includes a novel, LCP-based assay that allows the immobilization and functional characterization of this membrane protein in a (curved) lipid bilayer. The results are helpful in linking functional to structural insight.

The reported findings are of high relevance to researchers in the immediate field and beyond, and are clearly original and novel. The work is of very high technical quality. The biggest problem is the illustration and paper writing. The manuscript is cumbersome to read, very verbose, repetitive, and requires significant revision to make it readable. Many of the figures are too small, hard to understand, and also require some revision. Once the writing and illustration is improved, this will make for a great Nat. Commun. paper.

Specific comments for improvement:

1.) Improve structural representations in Figure 3, 4, 5 and Supplementary Figure 5 for easier understanding.

2.) Several sentences throughout the manuscript need rephrasing in order to be understandable, e.g. Page 16: '(c) The positive patch speculated for Pi release. (d) Superimpose of lysoPA to the aaPlsYPi structure.'

3.) Several statements are repeated multiple times, making the manuscript verbose rather than concise. Examples include the repeated comparison of the predicted 5 TM segments compared to the found 7 TM helices, the center paragraph of p. 20, multiple mention of the Asp-His dyad absence, and so on.

4.) (minor) Page 2: PlsY is not strictly speaking an orphan enzyme since its sequence has been described previously.

5.) (minor) Page 3: Michaelis-Menten complexes is not a commonly used expression in this context. Maybe use 'enzyme-substrate' or 'enzyme-product complex'.

6.) (minor) Page 6/7: Compare results from kinetic measurements (specific activity/ K_m) with previously published data (e.g. Lu et al., 2007).

Response to Reviewer's Comments:

Reviewer #1 (Remarks to the Author):

This paper reports the structure of PlsY. The data are comprehensive and convincing. The conclusions reveal a unique catalytic mechanism for an essential enzyme in Gram-positive pathogens. The results and conclusions are clearly presented.

We thank the Reviewer for the favorable comments.

There are a couple of places the presentation could be improved,

We have improved some of the figures (Fig. 2c, Fig. 3, Fig. 4, Fig. 5 and Fig. S5) as suggested by the Reviewer #3. We have also made efforts in correcting grammar errors.

and the impact of the work would improve by discussion of the structural biology of the three step pathway that is conserved in pathogens. Is there any implications from the structures that suggest a deeper understanding of phospholipid synthesis as a process?

The structure aspect of the biosynthesis pathway of phosphatidic acid is discussed in the revised manuscript.

The authors should consider making comments about the recently published structure of PlsC (Nat. Struct. Biol. 24:666-671, 2017). This is the next enzyme in the PlsX/PlsY/PlsC pathway, and the PlsY structure completes the structural overview of the pathway. PlsX was determined previously. Now that there is a structural view of the entire pathway to phosphatidic acid in pathogens, is there any deeper understanding of how the pathway works as an integrated process that can be gleaned by examining the structures.

A brief discussion of the PlsC structure in the context of the PlsX/PlsY/PlsC pathway has been added to the revised manuscript.

With respect to the pathway for acylP to find the active site, do the authors think that acylP must first be placed into the membrane and diffuse into the active site? AcylP is produced by PlsX, and another way of asking the question is: Do the structures suggest release of acylP into the membrane or is it possible that PlsX directly hands off its product. Likewise, is there any indication from the structures that PlsY and PlsC would form a complex, or do the authors interpret their structures indicate that the substrates diffuse out of one enzyme into the membrane, where they find the next step in the pathway. Exogenous fatty acids are activated to acylP by their phosphorylation on a fatty acid binding protein. Could the acyl chain be handed off in this case, or is the acylP first deposited into the membrane?

We thank the Reviewer in raising this interesting topic for discussion. We feel the incorporation of the relevant discussion should improve the manuscript and enhance its general interest.

We hypothesize that substrate channeling occurs between PlsY and the acylIP synthesizing proteins, namely PlsX and FakB, based on the fact that acylIP is very unstable, especially under temperatures for the growth of hyperthermophiles. Supporting bioinformatics evidence show the existence of fused genes of PlsY and FakB in two thermophiles. Similar substrate channeling mechanisms might also apply to PlsY and PlsC because lysoPA is potentially toxic to cells, and the direct hand-over would prevent its accumulation in membranes. The hypothesis remains to be tested by experiments.

Not sure that calling PlsY an evolutionary orphan is the correct term. It may be a connection to the deep past when acylation reactions occurred before the emergence of acyl group carrier cofactors. But an orphan?

We thank the Reviewer, as well as the Reviewer #3 in pointing out the incorrect term. The word 'orphan' has been removed.

Fig. S2a. What was the solvent for the gel filtration chromatography. Could not find it in the methods.

The solvent used for gel filtration in Fig. S2a was referred as Buffer D in the 4th paragraph under the Method section. It contains 0.03% (w/v) DDM, 150 mM NaCl, 20 mM Tris-HCl pH 8.0.

The text refers to this figure as a reconstitution of PlsY with LCP. Please clarify.

Fig. S2a and Fig. 2b were meant to be referred as 'purified aaPlsY'. This has been clarified as the following:

“For the coupled assay, purified aaPlsY (Supplementary Fig. 2a, 2b) and acylIP were reconstituted into LCP which was then deposited on the sidewall of microplate wells to avoid interference for later spectroscopic measurements.”

Reviewer #2 (Remarks to the Author):

The manuscript describes structural and functional studies of the enzyme glycerol 3-phosphate acyltransferase PlsY from *Aquifex aeolicus* (aaPlsY). A collection of five crystal structures are presented with different combinations of bound substrates and products, providing a framework for the complete description of substrate binding and catalysis in this enzyme class. Importantly, an enzymatic assay developed to enable monitoring of catalysis in lipidic cubic phase mixtures, also used for crystallization of the enzyme, supports the claim that the enzyme is active in this environment. The assay is used for extensive mutagenesis studies that support the claims made regarding substrate/product coordination and catalysis. Based on the crystal structures, the authors propose a novel enzymatic mechanism, that doesn't rely on an aspartate-histidine dyad involved in catalysis in other acyltransferase enzyme classes. The 'substrate-assisted catalysis' proposed involves a direct nucleophilic attack from the glycerol 3-phosphate hydroxyl to the carbonyl group of the acyl phosphate with proper coordination of the substrates through multiple interactions with enzyme residues. Finally, the authors provide a re-evaluation of previous findings in light of the crystal structures (topology, assignment of residue functional roles), and offer explanations as to the reasons previous findings/conclusions have been reached.

Overall, this is a very comprehensive and impressive work that conclusively characterizes all functional aspects of this class of acyltransferase enzymes. The work will be of great interest to structural biologists and biochemists alike.

The manuscript is generally well-written and fairly easy to follow. The figures and tables are well-crafted (though see some suggestions below). The methods section provides enough detail for reproducing experiments independently.

We thank the Reviewer for these complimentary comments and for the specific suggestions for improving the manuscript.

Minor comments:

1) When comparing the fold of aaPlsY with other 7-TM proteins of known structure, the structure of bacterial diacylglycerol transferase (Lgt) has been omitted from the comparison.

We thank the Reviewer in bringing up the Lgt structure to our attention. We have added a panel in Fig. S3 to compare the topology and fold between Lgt and PlsY; and cited Lgt as another example with proposed lateral gates for lipidic substrate entrance and product exit.

Although the topology of Lgt is different compared to PlsY, the fold appears to be fairly similar based on visual inspection (PDB ID 5AZB). It is worth exploring the fold similarity and also commenting on any functional similarities since Lgt also accommodates lipidic substrates.

Regarding the fold, little resemblance between PlsY and Lgt were apparent to us. Therefore we did not go further in investigating structural similarities between the two.

2) It is mentioned that TMH4 and $\alpha 2$ act as dipoles to neutralize the phosphate charge of acylP. What are the characteristics of TMH4 and $\alpha 2$ that make them act as dipoles? Some clarifications on what makes them act as such would help with comprehension by the reader.

The following texts have been added to the legends of Fig. 4.

“The aggregate effect of individual backbone microdipoles aligned along the α -helices axis (black arrows) causes a dipole moment with its positive pole at the N-terminus and its negative pole at the C-terminus. The resultant partial positive charges (δ^+) at the N-termini of the $\alpha 2$ (n=10) and the TMH4 (n=13) neutralize the phosphate charges of 16:0-P.”

3) The number of independent measurements (n) to determine enzymatic activity of aaPlsY and associated mutants should be included in Fig. 4c and/or Table 2.

The activity data for the wild-type aaPlsY was obtained from six independent experiments.

We did not perform independent repeats for the activity measurements of mutants. Instead, mutants were assayed under eleven different G3P concentrations. The data points were fitted with the Michaelis-Menten equation to obtain V_{max} and K_m . The activity data of mutants in Fig. 4e represent the $V_{max} \pm$ standard error from the curve fitting. The same data were used in Table 2.

In the previous version, we state this in the caption of Fig. 4 as: “Data are represented as $V_{max} \pm$ error from the Michaelis-Menten fitting.”

We have now modified the legends in the revised manuscript to make it more clear:

“(e) Activity of the wild-type and mutant enzymes. The wild-type activity ($34.55 \pm 3.14 \mu\text{mol min}^{-1} \text{mg}^{-1}$, mean \pm standard deviation) was obtained from six independent experiments, and the mean value was set to 100%. Purified mutants were assayed under eleven different G3P concentrations without technical replicates. G3P concentrations typically span three orders of magnitude with the high concentrations saturating the system. The V_{max} and standard error from the Michaelis-Menten fitting are reported.”

A note has been added to Table 2: “See Fig. 4e for activity measurement of mutants.”

Similarly, the clarification of error bars in Fig. 1d-f, Fig. S2c, and Fig. 6d are provided in the revised manuscript.

In addition, we added three paragraphs (highlighted blue) under Section “PIsY Activity assay using Pi-biosensor” in the Method part to clarify independent repeats and technical replicates.

4) It is mentioned that lysoPA (the product of the reaction that also acts as an inhibitor) interacts with PIsY more extensively than G3P, and thus should have higher affinity for the enzyme than G3P. Then how is lysoPA cleared? Through competition with the substrate acylP? In general, clearance of Pi is discussed but not clearance of lysoPA.

We have added the following text to discuss the lysoPA clearance:

“The release of the product inhibitor is probably driven by high concentrations of the substrates acylP and G3P. In addition, the exiting inorganic phosphate along the Pi-release path might aid the lysoPA clearance by electro-repulsion, as well as by competing for the phosphate clamp site.”

Suggestions for improving the manuscript:

1. p.3: "...deletion of both PIsY and the acylP-synthesizing enzyme PIsX are not possible..." clarify (is it lethal?) and provide citation.

Yes, it was lethal when both PIsX and PIsY were deleted, according to the Reference 7.

2. p.6 Fig. 1c: It is very difficult to discern the products/bands in the thin layer chromatography assay. The contrast of this panel must be improved.

We agree with the Reviewer. The brightness and contrast of Fig. 1c have been adjusted with a note in the caption: “The brightness and contrast of the whole image have been adjusted to enhance the substrate and product bands.”

3. p.6: In text a lag phase of "approximately 2min" is mentioned, but in Fig. 1 legend "The 3-min lag phase" is mentioned. Which of the two is true?

The lag phase lasts for approximately 2 min depending on how quickly the first data point was read by the plate reader. For the data shown here, the lag phase was 2 min.

Also in the corresponding panel (Fig. 1b) a discernible lag phase is not readily apparent and perhaps should be labeled for clarity.

We have added an arrow and corresponding caption to indicate the lag phase in Fig. 1b.

4. p.6 Fig. 1f: Is there a reason the points were not fitted with a curve in the enzyme loading graph?

The data points were fitted in the revised manuscript.

5. p.8 Fig. 2c: The color-coding of the topology diagram is overwhelming. At the very least a color-coding inset should be included. Also consider simplifying, perhaps dropping the color-matching to TM segments in (a) and (b), since the topology diagram is already organized per transmembrane segment.

We thank the reviewer for the suggestion in making the figure simpler and less confusing. We have dropped all the colors except for the Cys-mutant markers used for the Substituted Cys-Accessibility Method from Reference 12.

6. p.10 Fig. 3a,b: These panels should be made somewhat bigger, as their small size requires increased focus to follow the points being made.

We have made Fig. 3a bigger.

Fig. 3b in the original submission redundantly shows the overlapping ligands in the active site (as in the original Fig. 3c), and is therefore removed.

7. p.31: "...to inactivate DeoD, which was experimentally confirmed." Add how inactivation was experimentally confirmed.

We appreciate the Reviewer's careful reading.

DeoD (a purine nucleoside phosphorylase) transfers free phosphate to 7-methylguanosine. This activity can be monitored by the fluorescence decrease of the Pi-biosensor. The heat-inactivation of DeoD was verified by this activity assay. Such details have been added in the revised manuscript.

8. p.42 Sup. Fig. 2: Improve contrast of panel (b) (SDS-PAGE)

Done.

9. p.43 Sup. Fig. 3: In (b) a label for the third (from left) panel is missing (presumably Bax inhibitor motif protein based on the main text)

Done.

10. The manuscript is well written but does infrequently contain some grammatical/syntactical errors. A quick scan from a copy editor will help to eliminate them.

We have gone through the manuscript carefully and made efforts in improving the writing.

Reviewer #3 (Remarks to the Author):

Li et al. report a structural and functional analysis of the bacterial glycerol-3-phosphate acyltransferase (GPAT) PlsY, an intergral membrane protein. This enzyme uses acyl-phosphate as an acyl donor rather than the commonly found substrates acyl-CoA or acyl-ACP. The structure and mechanism was previously unknown.

The authors now describe five high-resolution structures of a GPAT in multiple states and at high resolution, including substrate- and product-bound states. The enzyme does not seem to undergo large conformational changes during the reaction mechanism. The structures provide seminal and highly relevant insight the architecture of PlsY and provide important clues to its mechanism. Based on the structural insight and on extensive biochemical characterization, Li et al. propose a plausible and novel mechanism for the acylation reaction. The functional analysis includes a novel, LCP-based assay that allows the immobilization and functional characterization of this membrane protein in a (curved) lipid bilayer. The results are helpful in linking functional to structural insight.

The reported findings are of high relevance to researchers in the immediate field and beyond, and are clearly original and novel. The work is of very high technical quality. The biggest problem is the illustration and paper writing. The manuscript is cumbersome to read, very verbose, repetitive, and requires significant revision to make it readable. Many of the figures are too small, hard to understand, and also require some revision. Once the writing and illustration is improved, this will make for a great Nat. Commun. paper.

We sincerely thank the Reviewer for the overall positive comments and fair critique of our paper. We have revised the manuscript according to the specific suggestions below.

Specific comments for improvement:

1.) Improve structural representations in Figure 3, 4, 5 and Supplementary Figure 5 for easier understanding.

For Fig. 3, we have removed the redundant Fig. 3b and made the other three panels bigger. Along with other improvements, this figure is now easier to read.

For Fig. 4, we have re-arranged the panels to make more space for Fig. 4a, 4c, and 4d. The ribbon representations in Fig. 4a and 4d are removed in the revised figure because they obscure the interactions in the previous version. When appropriate, dashed lines indicating interactions are colored differently for better visualization.

For Fig. 5, Panel (a) and (b) are re-oriented for better display of the diffusion pathway. For the same purpose, the two gate-forming helices were colored differently, and the active site ligands are made more visible by thicker stick representations. To better visualize the interactions between PlsY and the products, Fig. 5d and 5e are modified the same way as Fig. 4a.

For Fig. S5, Panel (a) has been changed to thick tubes with color codes for easier tracking. In addition, a lysoPA molecule was added to the active site to orient readers. Panel (b) has also been modified for better clarity. The Ans37 that assumes different conformations in the two structures is highlighted as thicker sticks.

2.) Several sentences throughout the manuscript need rephrasing in order to be understandable, e.g. Page 16: '(c) The positive patch speculated for Pi release. (d) Superimpose of lysoPA to the aaPIsY_{Pi} structure.'

We have re-written the two sentences and revised our manuscript carefully for better readability.

3.) Several statements are repeated multiple times, making the manuscript verbose rather than concise. Examples include the repeated comparison of the predicted 5 TM segments compared to the found 7 TM helices, the center paragraph of p. 20, multiple mention of the Asp-His dyad absence, and so on.

The comparison of the TM helices between the structure model and the SCAM model is only discussed in the 'Discussion' section of the revised manuscript.

The center paragraph of p. 20 in the original manuscript (regarding the compact build of PIsY) has been removed.

The discussion about the mechanism has been made more concise.

In addition, we have revised our manuscript as the following:

We have removed the repetitive discussions about Gly102/Gly103 in the 'Results' section.

We have removed the paragraph for substrate access and product release in the 'Discussion' section.

4.) (minor) Page 2: PIsY is not strictly speaking an orphan enzyme since its sequence has been described previously.

The word 'orphan' is removed in the revised submission.

5.) (minor) Page 3: Michaelis-Menten complexes is not a commonly used expression in this context. Maybe use 'enzyme-substrate' or 'enzyme-product complex'.

Done.

6.) (minor) Page 6/7: Compare results from kinetic measurements (specific activity/ K_m) with previously published data (e.g. Lu et al., 2007).

The kinetic differences between the two studies are discussed under 'Discussion'.